# Fibre optic distributed acoustic sensing of volcanic events

Philippe Jousset [1✉], Gilda Currenti [2✉], Benjamin Schwarz [1], Athena Chalari [3], Frederik Tilmann [1,4], Thomas Reinsch [1,8], Luciano Zuccarello [5,6], Eugenio Privitera [2] & Charlotte M. Krawczyk [1,7]

Understanding physical processes prior to and during volcanic eruptions has improved significantly in recent years. However, uncertainties about subsurface structures distorting observed signals and undetected processes within the volcano prevent volcanologists to infer subtle triggering mechanisms of volcanic phenomena. Here, we demonstrate that distributed acoustic sensing (DAS) with optical fibres allows us to identify volcanic events remotely and image hidden near-surface volcanic structural features. We detect and characterize strain signals associated with explosions and locate their origin using a 2D-template matching between picked and theoretical wave arrival times. We find evidence for non-linear grain interactions in a scoria layer of spatially variable thickness. We demonstrate that wavefield separation allows us to incrementally investigate the ground response to various excitation mechanisms. We identify very small volcanic events, which we relate to fluid migration and degassing. Those results provide the basis for improved volcano monitoring and hazard assessment using DAS.

[1] GFZ, German Research Centre for Geosciences, Telegrafenberg, D-14473 Potsdam, Germany. [2] Istituto Nazionale di Geofisica e Vulcanologia, Osservatorio Etneo, Piazza Roma 2, Catania, Italy. [3] Silixa Ltd., Silixa House, 230 Centennial Park, Centennial Avenue, Elstree WD6 3SN, UK. [4] Institute for Geological Sciences, Freie Universität Berlin, Berlin, Germany. [5] Istituto Nazionale di Geofisica e Vulcanologia, Sezione di Pisa, Via Battisti 53, Pisa, Italy. [6] School of Environmental Sciences, University of Liverpool, 4 Brownlow Street, L69 3GP Liverpool, UK. [7] Institute for Applied Geosciences, Technical University Berlin, Ernst-Reuter-Platz 1, D-10587 Berlin, Germany. [8] Present address: Fraunhofer IEG, Fraunhofer Research Institution for Energy Infrastructures and Geothermal Systems IEG, Am Hochschulcampus 1 IEG, 44801 Bochum, Germany. ✉email: philippe.jousset@gfz-potsdam.de; gilda.currenti@ingv.it

About a tenth of the world's population lives within the potential footprint of volcanic hazards, and volcanic eruptions regularly claim lives, damage properties and can cause major disruption to air traffic[1]. Multiparametric observations of volcanoes can improve our understanding of volcanic processes, and combined studies and integrated interpretation have been successful for issuing timely warnings and saving lives[2]. Volcano seismology has been efficient for describing the elastic and attenuation structure of volcanic edifices and model eruptive phenomena. For example, it can illuminate the geometry of the volcanic plumbing system and provide information on the location of magma bodies and hydrothermal systems at depth[3–6]. One central goal of volcano seismology[3] is to describe the characteristics and understand the nature of the seismic signals in association with magma migration and hydrothermal fluid flow circulation in the edifice. Volcanic processes generate a large variety of seismic signals (Supplementary Note 1), such as volcano-tectonic events (VT, 3–40 Hz), long period events (LP, 0.2–5 Hz), very-long period signals (VLP, 0.05–0.2 Hz), tremor (~0.1–10 Hz), and volcano-explosive signals (VEQ, explosion quakes, ~1–10 Hz). For example, different models have been proposed to explain the source mechanism of LP events and tremor: most researchers attribute their origin to the complex interplay between magmatic-hydrothermal fluids and their hosting rocks[7] but alternative hypotheses such as slow-rupture processes[8] have also been proposed. Volcanic explosions produce energy propagating both in the subsurface as seismic waves and in the atmosphere as acoustic waves. In recent years, infrasonic data has been used to complement seismic records[9] to gain further insights into the nature of acoustic sources at volcanoes, such as the explosion location and its energy release[10–12], contributing to the improvement of our understanding of fundamental eruption source parameters[13–17]. In particular, the analysis of the partitioning of acoustic and seismic energy during explosive eruptions reveals changing conditions within the conduit[18].

The quantitative analysis of acoustic and seismic signals associated with volcanic events is therefore a fundamental step towards shedding light on the dynamics of volcanic processes, for an improved assessment of volcano unrest[3]. Clearly, denser deployments of seismic sensors[19] lead to the detection of smaller events[20,21] providing more detailed seismic tomographic images. Although fundamental eruption processes are understood and basic precursory signals prior to eruptions can generally be detected, incomplete structural knowledge and the inability of current monitoring networks to detect small but possibly significant signals prevent volcanologists from accurately describing subtle, yet decisive fundamental volcanic processes[22].

Here, we extend our capability and sensitivity of deciphering volcanic phenomena by discriminating tiny volcanic events within the volcanic tremor and by determining the local subsurface structure at ever higher resolution, which provides us with insights into the non-linear response of volcanic rocks. We record seismo-acoustic waves from volcanic activity at Etna volcano (Italy) by distributed acoustic sensing (DAS)[23–26] using a dedicated fibre optic cable deployed at safe distance from the active craters. Etna (Fig. 1, Supplementary Note 2) is the largest, most active and most tourist-visited volcano in Europe, in whose vicinity more than 1 million people live. Etna volcanic activity is characterized by frequent effusive and explosive eruptions[27]. One of the significant eruptions at Etna[28] (December 2018) serves as a reminder of the ever-present hazard associated with e.g., lava flows, ash fallout, and earthquakes. Etna has been extensively studied[27] and is densely monitored with state-of-the-art instrumentation (Supplementary Note 2). From 30th August to 16th September 2018, we connected an iDAS interrogator ("intelligent Distributed Acoustic Sensing"[29]) to a > 1.3 km long standard telecommunication multimode fibre optic cable (Fig. 1) buried at about 15–25 cm depth in scoria deposits at ~2–2.5 km distance from the 5 active craters at the summit of Etna volcano (Method: DAS, fibre and conventional sensors, Supplementary Fig. 1). We recorded densely distributed (every 2 m) dynamic strain rate signals associated with weak Etna activity (Supplementary Fig. 2, Supplementary Tables 1 and 2), e.g., volcanic explosions, small volcanic transients (degassing), local volcano-tectonic earthquakes, as well as with atmospheric phenomena including hail and thunderstorms. We validate the DAS strain rate measurements with strain rate estimates from broadband seismometers and geophones (Method: DAS Strain rate and strain validation, Supplementary Fig. 3 and 4, Supplementary Table 3), and compare to signals from infrasound sensor arrays. Standard volcano-seismology analysis, wave and strain propagation modelling tools, and techniques such as wave-field separation and reconstruction[30] make it possible to quantify hidden subsurface structural features and accurately detect and locate volcanic events. We find evidence for non-linear interactions of acoustic waves with the near-surface scoria deposits, triggering resonance phenomena in the subsurface, allowing us to estimate the thickness of the scoria layer. In addition, clear reflection signals of the acoustic induced waves allow us to identify a superficial reflector hidden below the scoria layer and quantify its azimuth. Finally, we detect and identify tiny volcanic events hidden within the tremor and tentatively interpret them as surface degassing and internal fluid motion by analysing, e.g., their inter event times[31,32].

## Results

**Volcanic explosions image the subsurface.** We infer the subsurface structure and its non-linear response to seismo-acoustic signals generated by volcanic explosions. Explosions occurred at New South East Crater (NSEC) on 5th September 2018 (Supplementary Table 1) with blast sounds, which were audible by residents living on the eastern flank of Etna volcano (Supplementary Note 2). We focus on the explosion at 10:54:11 UTC (Supplementary Movie 1). The initial supersonic shock waves (close to the source) propagate further as sonic acoustic waves[13]. Acoustic waves generally decay with the inverse of the traveled distance due to geometrical spreading, except close to the source. Our infrasound array (CARB, 2.3 km away from NSEC, Figs. 1, 2c, f) recorded a pressure pulse with a maximum positive amplitude of 107 Pa (or 140 Pa peak-to-peak) yielding[13] an explosion energy of $\sim 2.5 \times 10^{11}$ J. The energy generated by the explosion propagates as a shock wave no further than ~100 m from its source, a distance defined by the radius at which the transition between supersonic to sonic (Mach number of 1) occur[13], and then decays as an acoustic pressure wave following an inverse proportionality relation with distance[13] until reaching the instruments. In the DAS records, we observe and analyse two strain rate sequences, allowing us to infer the subsurface structure and explain ground response features.

The first sequence comprises a ~4 s long signal (~1–10 Hz, "low frequency") corresponding to the complex seismic wave field (mostly Rayleigh waves) induced by the explosion (Fig. 2). The branch B2 of the cable (Fig. 1) is aligned with the radial direction with respect to NSEC. As DAS measures strain rate along the fibre direction, we expect larger amplitudes along B2 than along B1. Instead, larger amplitudes occur on branch B1. In addition, the observed strain rate is amplified (channels 315–340) in the proximity of a fault zone[33] (FZ, Fig. 1). Using 3D wave propagation simulations, we illustrate how Etna topography and volcanic structures (a fault-zone and tomographic models[34]) may explain the wave-field variability (Supplementary Figs. 5, 6 and 7,

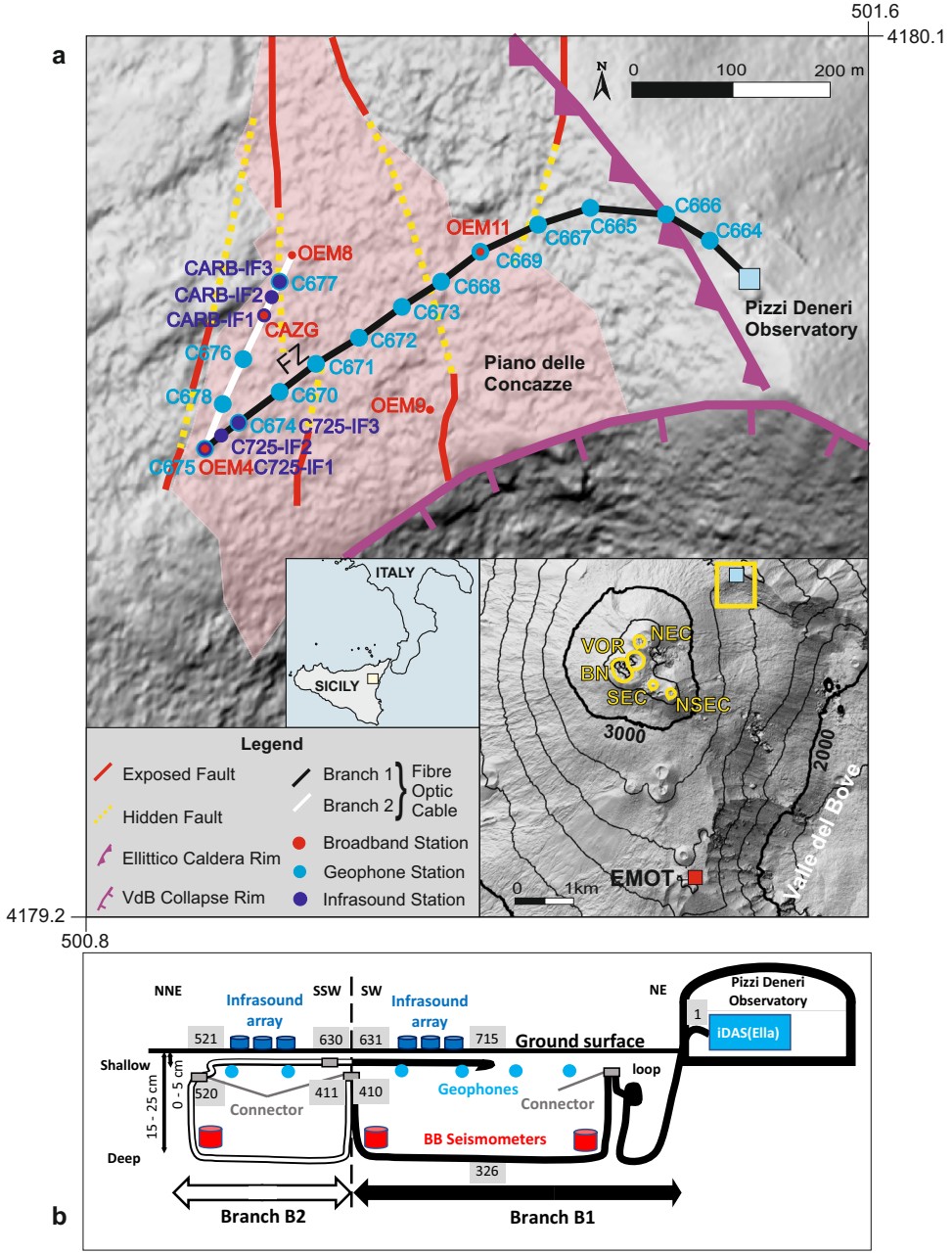

**Fig. 1 Fibre optic cable, seismometer and infrasound sensor locations and deployment near Etna volcano summit (Piano delle Concazze) and Valle del Bove on the digital elevation model[58]. a** The iDAS interrogator (Method: DAS, optical fibre and conventional sensors), set up at Pizzi Deneri Observatory (light blue square), is connected to the fibre indicated by the black ("branch B1") and the white ("branch B2") lines, respectively. **b** Sketch of the cable deployment. From the interrogator (inside and around the observatory, channel 1–50), the cable is buried in compacted material (channels 50 until 200) and then in lose scoria deposits[33] (transparent reddish area in a.), at about 15–25 cm depth (deep section) along B1 with channels 1 to 410, then the cable turns (still within the deep section) along B2 with channels 411 until 520, then the cable has a shallow section (under a few cm of scoria and lying directly above the deep cable), from channels 521 until 630 (with same geographic location as deep channels 520 until 411, respectively), and finally, the shallow cable turns along B1 (still above the deep cable) from channels 631 until 715 (with same geographic location as deep channels 410 until 326). Insets: Local and regional contexts. Summit craters' locations: NSEC (New South-East Crater); SEC (South East Crater); BN (Bocca Nuova); VOR (Voragine); NEC (North-East Crater). Red square: Thermal camera location: EMOT. The yellow box indicates the location of the main map.

respectively, and Method: 3D wave propagation modelling). Models which include the fault zone indicate clear strain rate amplification near the fault area, whereas velocity waveforms do not change significantly, in agreement with geophone observations. Synthetic travel times and amplitudes cannot be explained by laterally homogeneous models. In contrast, models with three-dimensional tomographic models[34] and on which a shallow low velocity layer following the topography is added, simulate strain

rate amplitudes along branch B2 to be slightly lower than amplitudes along branch B1.

The second sequence, superimposed onto the first sequence, comprises a ~2 s long signal (~16–21 Hz, "high-frequency") induced by the acoustic wave from the explosion, whose arrival time is in excellent correspondence with the recorded infrasound signal (acoustic pulse). Unlike the first-arriving low frequency signal, amplitudes of the high frequency signal are similar all

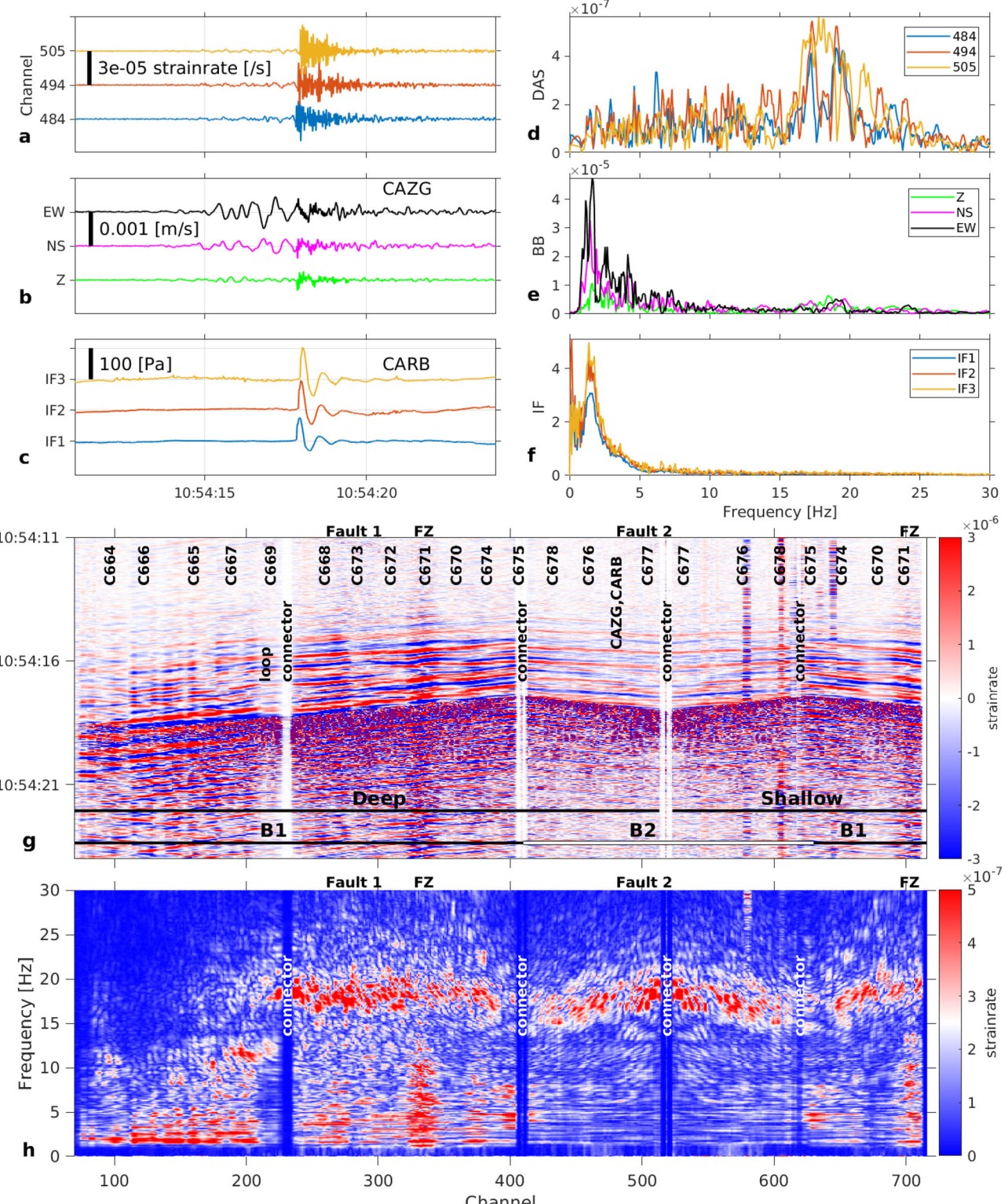

**Fig. 2 Explosion at Etna New South-East Crater (NSEC), September 5, 2018, at 10:54:11. a** Strain rate from distributed acoustic sensing (DAS) records at channels 484 (blue), 494 (red) and 505 (yellow), corresponding to positions of infrasound sensors in (**c**). Fibre channel position accuracy ±3 m (Method: DAS interrogator, fibre optic cable and conventional sensor network characteristics). **b**. Velocity seismograms from broadband seismometer CAZG (Supplementary Table 3), near DAS channel 494. **c** Pressure records from infrasound sensors CARB-IF1, 2, 3. **d** Strain rate (**a**) spectra. **e** Ground velocity (**b**) spectra. **f** Pressure (**c**) spectra. **g** Strain rate record at the 710 DAS channels along the 1.3 km fibre around the explosion time. B1 and B2 are the two geographically distinct branches in Fig. 1. FZ: fault zone (~50 m width), at channels 315–340 (deep cable) and channels >700 (shallow cable). **h** Strain rate-frequency distribution along the cable. Note higher strain rate amplitudes at low frequencies 1–10 Hz (seismic signal) for branch B1 and at high frequencies 18–21 Hz (infrasound induced signal) for both branches.

along the cable, except at the initial part of branch B1, close to the observatory (channels < 200; Fig. 2h).

Surprisingly, the high frequency signal, detected by both DAS and seismometers, is not recorded by the infrasound sensors, although their instrumental response[35] extends to at least 200 Hz (Supplementary Fig. 8, Supplementary Note 3, Supplementary Movie 2). At locations where the scoria layer is very thin or even inexistent, the high frequencies are absent from both the DAS records (channels 1-200) and the velocity seismograms (geophones C664, C665 and C666, Supplementary Fig. 9). This discrepancy implies that the high frequency signal results from interactions between the infrasound wave and the scoria layer (made of pyroclastic grains of 1–3 cm diameter, Supplementary Note 4) deposited by previous eruptions over the competent/compacted rock substratum[33]. The infrasound frequencies (<2 Hz) shown in Fig. 2f do not show up on the cable (Fig. 2h) laid in the scoria area, but are present in the compacted substratum (traces 100–200). When hit by the acoustic wave, each scoria particle interacts with its neighbours, resulting in the resonance of the whole scoria layer above the competent substratum. The infrasound signal in the DAS record results therefore from the non-linear response of the scoria layer, rather than directly from cable-air coupling.

This explosion (with pressure peak ~107 Pa) induced a maximum strain rate amplitude of ~$5 \times 10^{-5}$ s$^{-1}$ (Fig. 2). However, not all explosions recorded during our experiment triggered the high frequencies. Another explosion at NSEC on the same day at 14:04 associated with an infrasound pulse of 23.5 Pa generated both seismic and infrasound sequences (Fig. 3); however, the maximum strain rate amplitude at all frequencies is $2 \times 10^{-6}$ s$^{-1}$, which is only a fifth of the amplitude expected from a linear pressure-strain relation ($11 \times 10^{-6}$ s$^{-1}$). Note that also the seismic records (Fig. 3b, e) do not show high frequencies for this explosion. In addition, two infrasound events with peaks 2.7 Pa and 4.8 Pa on 15.09.2018 at 02:31 and 16.09.2018 at 09:26 should have generated strain rate signals with amplitudes of about $1.3 \times 10^{-6}$ s$^{-1}$ and $2.2 \times 10^{-6}$ s$^{-1}$, respectively. We observe no amplitude changes above the noise floor (~$10^{-8}$/s) at the resonance frequencies (16–21 Hz). Therefore, resonance phenomena appear to occur only when the pressure amplitude (high strain) is sufficiently high or the pressure change sufficiently fast (high strain rate) to excite the non-linear soil response at the resonance frequency. Under high strain rates loading[36], the soil response comprises complex effects (e.g., friction laws[37]) within the scoria deposits made of a solid matrix (rock grains), gas (air) and liquid (meteoric water).

We observe a spatial shift in the resonance frequency along branch B2 (16 Hz and 21 Hz at channels 425 and 510, respectively), which we attribute to a variable thickness of the scoria layer along the cable profile. We relate the resonance frequency $f$ to the scoria deposit thickness $h$ by[38] $f = \frac{Vs}{4h}$, $Vs$ being the shear wave velocity. Conventional seismic methods (Method: Ground velocity estimations and Supplementary Figs. 10 and 11) provide $Vs$ estimates ranging 400–1100 ms$^{-1}$ (apparent velocities) and 200–600 ms$^{-1}$ (from MASW analysis) yielding estimates for a scoria layer total thickness between 2.5 m ($f = 21$ Hz, $Vs = 200$ m s$^{-1}$) and 17 m ($f = 16$ Hz, $Vs = 1100$ m s$^{-1}$).

**Resolving the volcanic wavefield with DAS.** We resolve the seismo-acoustic wavefield in order to highlight signals from distinct volcanic processes and structures. The fibre optic cable, in a single measurement, records wavefields over a wide frequency range from distinct physical processes. In order to fully assess the rich information content of the DAS recordings, we perform wavefield separation based on a coherent wavefield subtraction scheme, recently developed for weak-wavefield imaging in controlled-source seismology[30] (Method: Coherent wavefield separation and data enhancement). The dense spatial sampling along the fibre enables us to separate interfering signals by first picking out and enhancing specific coherent components in slowness-distance-time space, and then subtracting them to make weaker contributions more visible. Proceeding in an iterative fashion, it is possible to reduce the noise level and highlight subtle signals hardly visible in the original record. We use these enhanced constituent wavefields to locate the source of volcanic explosions, identify small structures and estimate a local 1D velocity model.

Wavefield separation applied to the explosion records enhances the induced acoustic wavefield (Fig. 4), which facilitates automatic picking of the acoustic arrival times (Fig. 4c). We locate the event using differences between picked acoustic arrival times at different channels in a least-squares beamforming procedure[39] which provides a back-azimuth (~201°) pointing to NSEC. Concurrently, by applying a 2D template matching, we compare theoretical and picked acoustic arrival times in terms of RMS and semblance measures. The best fit is achieved when assuming NSEC as the explosion source location. As source and receiver are close (~2.2 km apart), we assume straight ray paths[40] and measure an apparent velocity of $355 \pm 13$ ms$^{-1}$ for the primary incoming acoustic wavefield (azimuth ~20°) along branch B2 (azimuth 24°). We observe a weak infrasound-induced back-propagating signal (channel 490, Figs. 1 and 3b) with an apparent velocity ~$432 \pm 17$ ms$^{-1}$, most likely corresponding to the reflection of the primary wavefield off a planar local structure. We investigate contributions of possible structures under the scoria deposits such as hidden faults[41], a local magnetic anomaly[42] and a local strain map[43]. A probable reflector has an azimuth of 145° ± 5° (Supplementary Fig. 12), in correspondence to a local magnetic contrast[42], interpreted as the border of a massive lava flow from Ellittico activity.

The enhanced coherent wavefield further allows us to more reliably estimate the 1D velocity model of the sub-surface (scoria and substratum). We obtain multimodal Rayleigh wave dispersion curves from multichannel analysis of surface waves (MASW) generated by jumps (Supplementary Fig. 11) performed along the cable. Our dispersion analyses allow us to retrieve a large number of modes, pointing to a strongly dispersive media due to many superposed volcanic strata with different velocities, in agreement with a succession of lava flows and scoria deposits. Inversions[44] of those dispersion curves provide vertical profiles of layers (first layer with shear-wave velocity of 200 ms$^{-1}$ and 3–5 m thickness; deeper layers have velocities up to 600 ms$^{-1}$ at about 20–25 m depth). Those thicknesses confirm our estimates (2–17 m) from the scoria deposit resonance frequency excited by the explosion. Indeed, derived shear wave velocities from MASW are lower than the estimated apparent velocities from the inter-channel travel time method (Supplementary Fig. 10). Using the range of shear-wave velocity (200–600 ms$^{-1}$), their corresponding depth (2–9 m) from the MASW analysis and the observed resonance frequencies (Fig. 2), we estimate[38] ($h = \frac{Vs}{4f}$) that only layers between the surface and maximum 4–5 m (shear wave velocities <300 ms$^{-1}$) are the ones most probably involved in the resonance due to the explosion. Generally, we can identify the dispersion modes clearly for jumps where the scoria layer is present, due to the marked contrast in shear velocity between the scoria layer and the substratum. Where the substratum is exposed at the surface, the dispersion modes are much less clearly identifiable or inexistent. We also note less clear dispersion curves, when profiles cross faults.

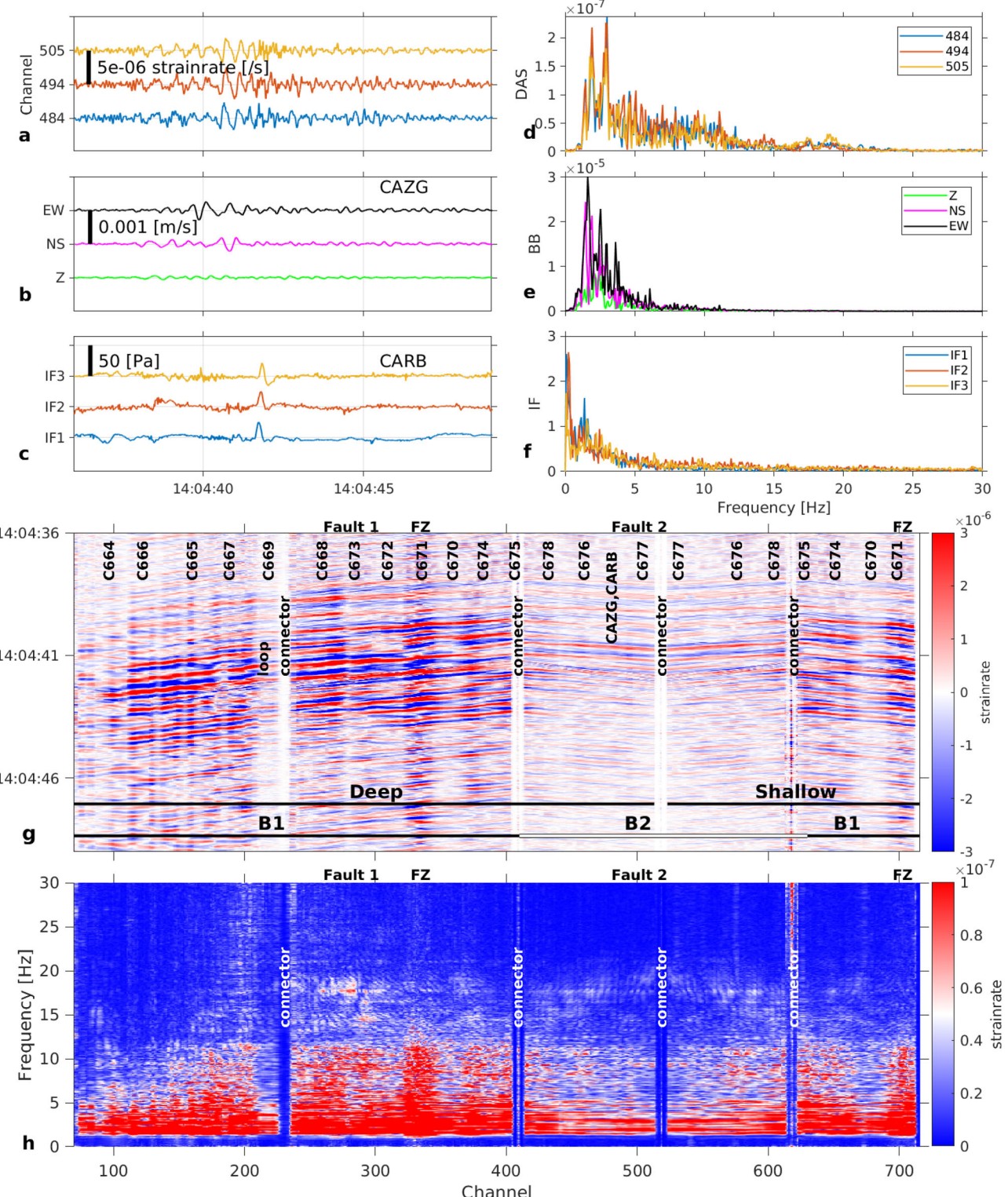

**Fig. 3 Explosion at Etna New South-East Crater (NSEC), September 5, 2018, at 14:04:35. a** Strain rate from distributed acoustic sensing (DAS) records at channels 484 (blue), 494 (red) and 505 (yellow), corresponding to positions of infrasound sensors in (**c**). Fibre channel position accuracy ±3 m (Method: DAS interrogator, fibre optic cable and conventional sensor network characteristics). **b** Velocity seismograms from broadband seismometer CAZG (Supplementary Table 3), near DAS channel 494. **c** Pressure records from infrasound sensors CARB-IF1, 2, 3. **d** Strain rate (**a**) spectra. **e** Ground velocity (**b**) spectra. **f** Pressure (**c**) spectra. **g** Strain rate record at the 710 DAS channels along the 1.3 km fibre around the explosion time. B1 and B2 are the two geographically distinct branches in Fig. 1. FZ: fault zone (~50 m width), at channels 315–340 (deep cable) and channels >700 (shallow cable). **h** Strain rate-frequency distribution along the cable.

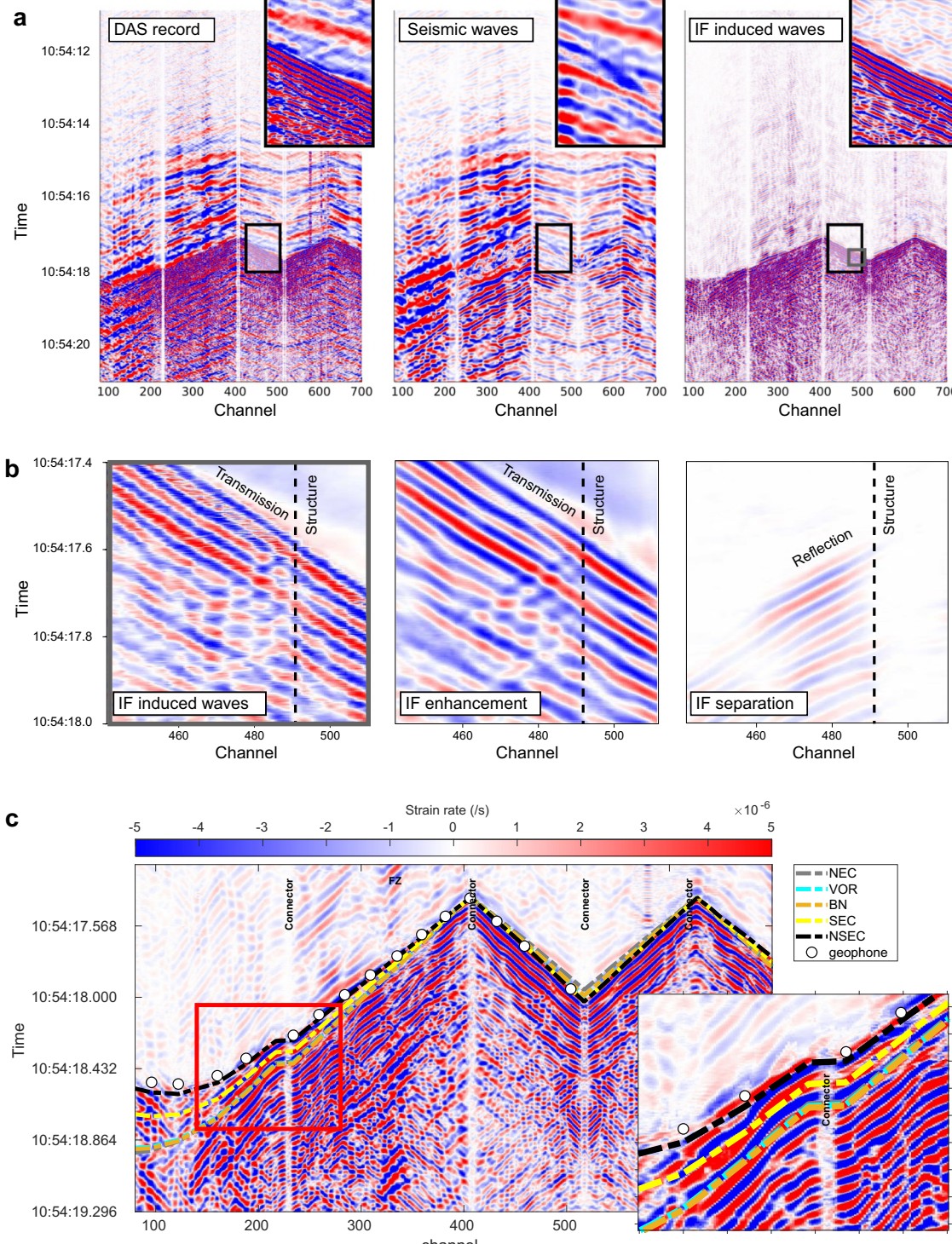

**Fig. 4 Coherent wavefield enhancement and separation for volcanic explosion record shown in Fig. 2. a** Stage 1: Separation of seismic and infrasound wavefield. Left: Original DAS records. Centre: Estimated contribution from seismic wave propagation. Right: Estimated contribution from infrasound induced wave propagation. The black frame indicates the closeup displayed in the top right corner of each image. The grey frame indicates the zoom-in of (**b**). **b** Stage 2: Separation of forward and backward propagating wavefield: (Left) Closeup of the DAS-infrasound wavefield (indicated by the grey frame in **a**). (Centre) coherence-enhanced infrasound wavefield. (Right) backpropagating energy stemming from a structure crossing the cable near channel 490, whose reflection properties (e.g., amplitude variations) can be more accurately delineated in the separated domain. **c** The observed strain rate arrival times are compared with theoretical arrival times for all craters, assuming an acoustic wave velocity[71] of 340 ms$^{-1}$ in the air (coloured lines for the craters; Lines for NEC, VOR and BN are hardly distinguishable as they nearly overlap). This 2D template matching is consistent with an explosion at NSEC (as verified in Supplementary Movie 1). White circles indicate the observed arrival times of the infrasound high frequency signal picked from the geophone records (Supplementary Fig. 9).

**Towards DAS volcano monitoring**. We show that monitoring of volcanic events can successfully be performed with DAS data, thanks to its high information content. The DAS strain rate data reveal frequent transients (~30–35 events per hour, each 5–15 s long, with amplitude ~$5 \times 10^{-7}$ s$^{-1}$, Fig. 5), barely visible in our seismic array data. In order to infer their characteristics, we apply three detection methods (Method: Detection methods for continuous monitoring) to two weeks of data (Supplementary Fig. 2). These methods detect many events (e.g., Fig. 5), however, events are not all detected by all methods. We find two types of transient events (Fig. 6): (i) STP (Single Tremor Pulses, 0.1–6 Hz), with strong coherence of DAS signals among channels, are better detected with the similarity method; (ii) DG events (Degassing events, 1–15 Hz), with low coherence between channels, are better detected with STA-LTA or stacking methods. DG and STP events are not associated with detectable infrasound signals, which is consistent with a low level of volcanic activity, and could only be identified with the DAS records (Fig. 7). Note that in the fault zone, STP and DG event signatures are different, due to the higher frequency content of DG events. DG events are likely associated with small and shallow intra-crater events, such as the one observed from the crater rim at NEC (Supplementary Movie 3). In contrast with the DG events, the strong coherence of STP waveforms among channels (including in the fault zone) suggest that STPs are generated at larger depth, although we cannot infer an accurate location with our spatially limited network.

Analyses of the inter-event time distributions of seismic events give hints on their physical mechanisms[31,32]. In general, inter-event time distributions of seismic activity are well approximated by the gamma distribution[31]:

$$P(\tau) = C\tau^{\gamma-1}e^{-\tau/a} \qquad (1)$$

where $C = \left[a^{\gamma}\Gamma(\gamma)\right]^{-1}$, $\Gamma$ is the gamma function, $a$ is the scale parameter, $\gamma$ is the shape parameter and $\tau$ is the normalized inter-event time obtained by multiplying the observed inter-event times by the average event rate $R$. A "universal" scaling law for tectonic earthquakes[31] has been proposed where $C = 0.5 \pm 0.1$, $\gamma = 0.67 \pm 0.05$, $a = 1.58 \pm 0.15$. The inter-event times of volcano-tectonic earthquakes at Etna has been found to follow the gamma distribution only during quiet periods[31], but not when magmatic stresses overcome the tectonic regime. In addition, the inter-event time distribution for Long-Period volcanic events at Etna has been found to deviate significantly from the "universal" scaling law, expected for tectonic activity[32].

We explore the inter-event time distributions of the transient events found by the three detection methods (Method: Detection methods for continuous monitoring). We find that the inter-event times are well approximated by gamma distributions (Fig. 5b). After rescaling[31] with the average event rate R, the probability density function of observed inter-event times deviates significantly from the typical gamma distribution expected for tectonic activity[31], although Etna was quiet during that period. In contrast, these distributions follow the gamma distribution found in 2005 for Etna LP events[32]. This suggests that the detected STP and DG events are most likely related to intermittent strain build-up and release similar to the source mechanism generating the LP events. Supplementary Movie 3 shows that DG events are the surface expression of fluid movement, i.e., pulsating degassing. As STP and DG events have similar waveforms for low frequencies (<1 Hz, Figs. 7 and 8, Supplementary Figs. 14 and 15), we propose that the STP events may be a signature of deeper fluid movement, powerful enough to be detected by the DAS array. Seismic tremor would be the combination of fluid pulses (e.g., gas bubbles) migrating within the conduit, with larger pulses

recorded as STP at depth and DG as pulses reach the surface. We emphasise that neither DG events (Fig. 7) nor STP events (Fig. 8) could be detected in the infrasonic data and barely identified in the records from seismometers at Piano delle Concazze, demonstrating the potential of DAS for continuous monitoring of small volcanic activity and identifying faint event characteristics.

## Discussion

Discovering hidden features of volcanic structures and understanding the ground response to volcanic processes would help deciphering complex signals to unravel eruption dynamics and precursors. Those objectives require dense and accurate geochemical, geophysical and structural observations. As demonstrated here, fibre optic cables interrogated with DAS technology offer a complementary tool for both characterization and quantitative assessment of volcanic structures and monitoring. We demonstrated that owing to the very high spatial sampling of DAS recordings, we are able to locate volcanic explosions, detect and characterize volcanic structural features including hidden volcanic features. In addition, we identified small transients within the persistent volcanic tremor, and showed they were at least of two types, namely STP and DG events. We tentatively interpret them as a signature of fluid migration within the plumbing system and pulsating degassing in the craters. In order to accurately locate these subtle signatures, more extended fibre-optic strain measurements are needed. Full appreciation of their origin and understanding their nature in relation to degassing and tremor will need to be integrated with additional volcanological observables, e.g., through thermal and ultraviolet imaging of volcanic degassing[45].

Complemented by infrasound records, further analysis of our DAS observations revealed a non-linear ground response of the scoria layer to incoming infrasonic pressure waves caused by volcanic explosions. This non-linear behaviour is likely to be observable at other explosive volcanoes and may also be triggered by atmospheric pressures sources such as thunderstorms (Supplementary Fig. 13). The coupling of the infrasound signal with the ground is of general interest in a wide range of cases, where explosions occur near the surface. The extensive distribution of intergranular cracks, dislocations and weak or failing grain contacts in poorly consolidated sediments may introduce non-linear behaviour of rocks[37,46,47]. Our findings open a path for quantifying soil properties by analysing the impact of external forcing from pressure sources, such as explosions. This analysis can lead to a better understanding of rock behaviour and slope stability.

One challenge of the application of DAS on active volcanoes may be the lack of available infrastructure. Dark-fibre telecommunication infrastructure has been shown to be particularly appealing and help reach acceptance and full appreciation of the actual affordability of DAS[23,24]. Once a cable with multiple fibres has been deployed, it can be used both for data transmission and interrogation with various optical sensing techniques. Thanks to its long-distance probing capabilities (currently at least several tens of kilometres), the interrogator can be set-up in a remote place, making fibre optic observations easier and safer than conventional sensor arrays, which need telemetry, on-site power supply and regular maintenance. Where possible, one or multiple fibre-optic cables deployed from the volcano's summit to remote locations would provide unique opportunities to deepen the understanding of the ground response, including the estimation of path effects and aiming at a better understanding the origin of volcanic phenomena, e.g., by performing full-waveform source inversion[48]. In addition, the availability of fibre optic cable submarine infrastructure[24,49] close to volcanic islands is also of great

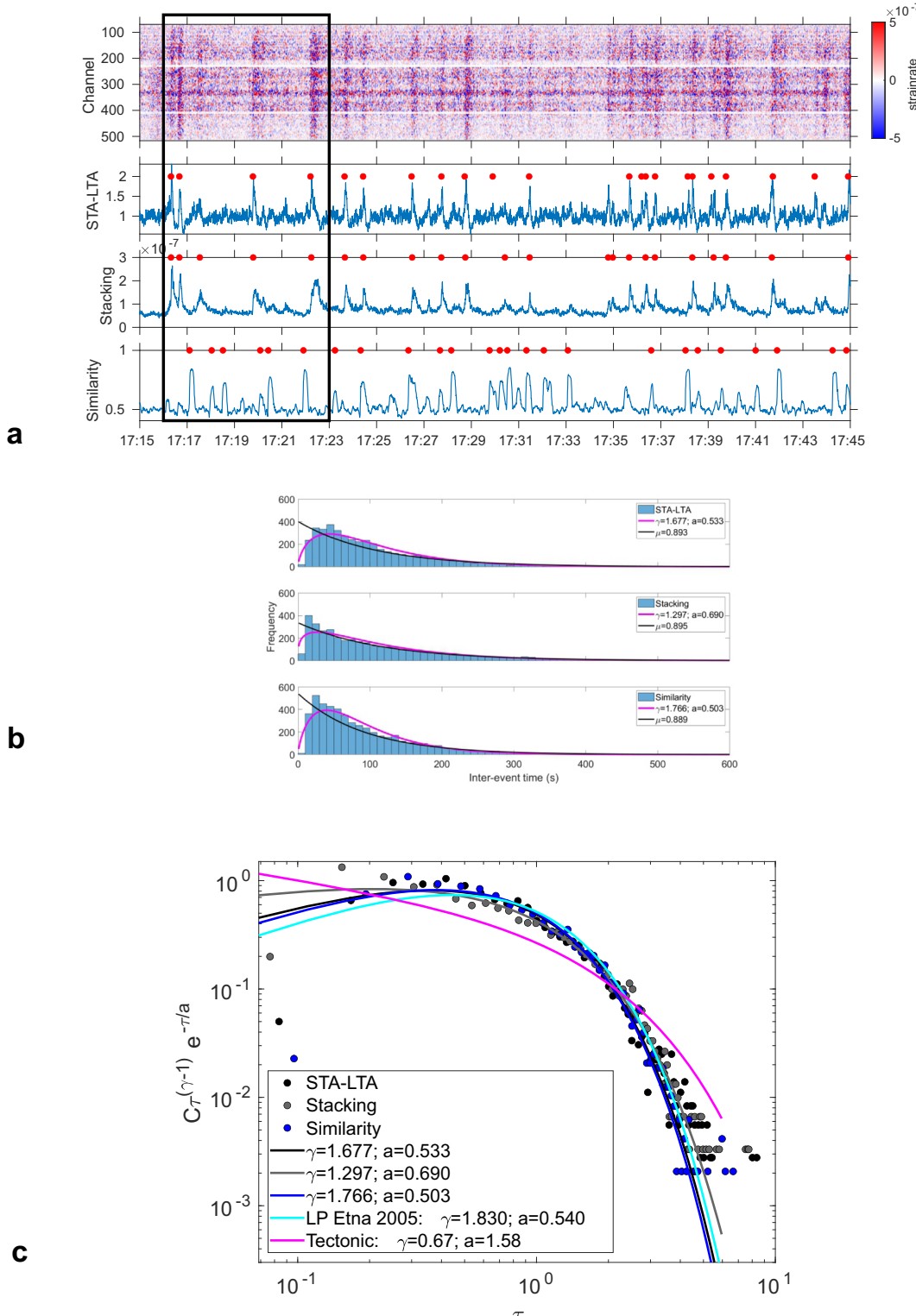

**Fig. 5 Continuous detection of weak volcanic events. a** Typical example of 30 min strain rate data (31/08/2018 17:15:00–17:45:00, filtered 0.1–5 Hz). 3 lower panels: detection results (red dots represent event detection times) based on (top) Short-term average (STA)–long-term average (LTA) with STA = 0.7 s, LTA = 10 s and threshold = 3; (middle) stacking (summation of trace amplitude); (bottom) local similarity algorithm. The black rectangle indicates the extend of Fig. 6a. **b** Histograms of inter-event times between detected events for the whole acquisition period (31/08/2018 until 16/09/2018, see all detections in Supplementary Fig. 2). For each detection method, the corresponding gamma distribution (pink) and exponential models (black) are given, with their parameters specified in the legend (top) STA-LTA (R ̴ 30.9 events/hour); (middle) stacking (R ̴ 28.3 events/hour); (bottom) local similarity (R ̴ 35.9 events/hour). **c** Distribution of observed inter-event time Δt of DAS detected events after rescaling by the average event rate R, i.e., τ = R Δt. Continuous lines represent fits of data to the theoretical universal gamma distribution (Method: Probability density functions of inter-event times), and are compared with gamma distributions for LP events at Etna[32] and tectonic events in Southern California[31].

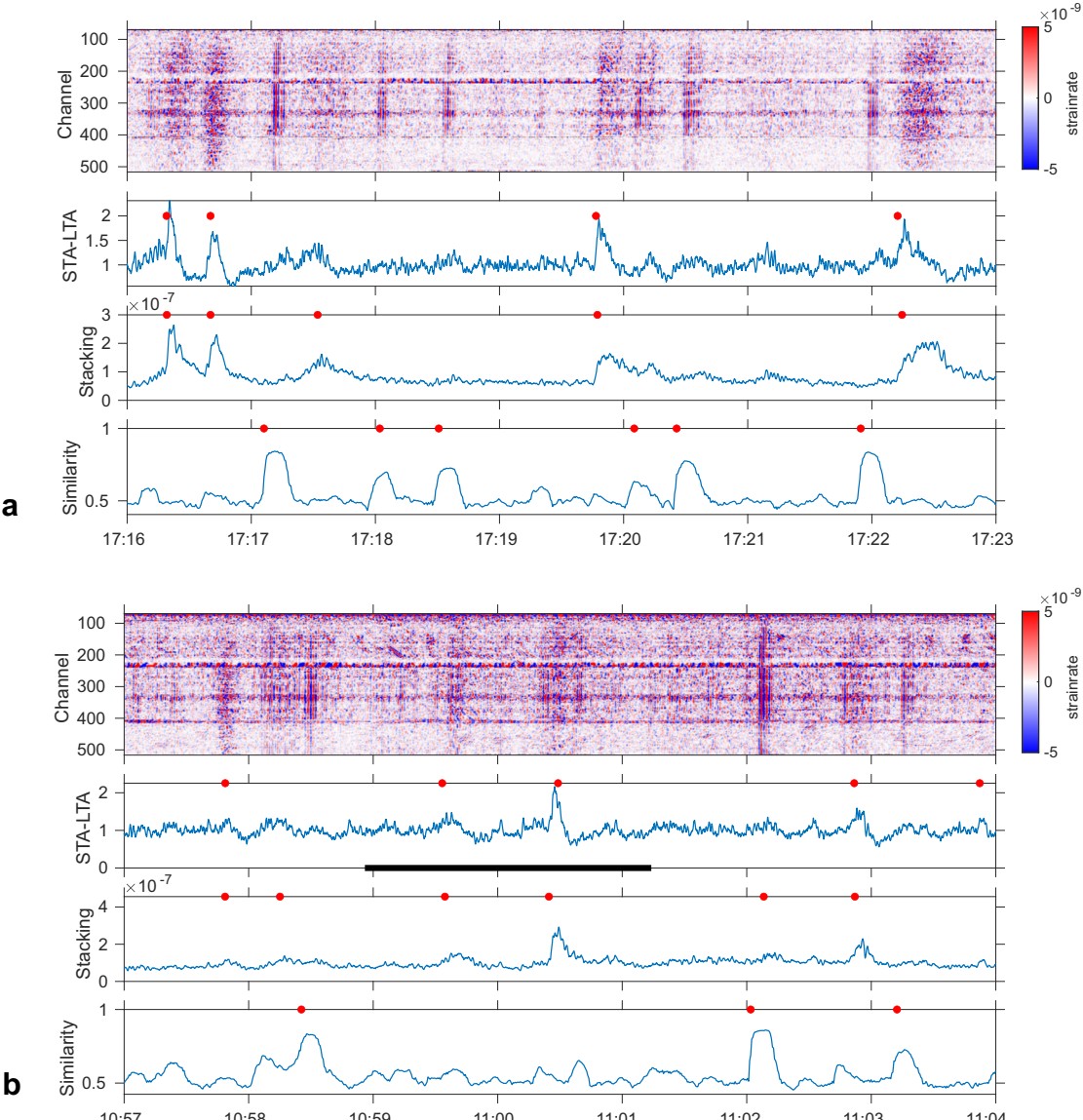

**Fig. 6 DAS detection of small transients.** Similar layout as Fig. 5a for few minutes of DAS records, except that strain rate (top panel) is plotted for DAS data filtered 0.1–0.6 Hz to highlight differences between transient patterns, e.g., those detected by the similarity method, i.e., STP events, from those detected by STA/LTA and the stacking method, i.e., DG events. STPs contain mostly low frequencies (1–2 Hz), whereas DG events have also higher frequency content (up to 10 Hz, see Supplementary Figs. 14 and 15). 3 lower panels: detection results (red dots represent event detection times) based on (top) STA-LTA (STA = 0.7 s; LTA = 10 s; threshold = 3); (middle) stacking (summation of trace amplitude); (bottom) local similarity. **a** Zoom (31/08/2018 at ~17:17) of Fig. 5a. **b** 12/09/2018 at ~11:00, during which a video was taken from North East Crater (NEC) rim (Supplementary Movie 3). Black line: time span of the video. Note that this event is detected with the STA-LTA and stacking detection methods, but not by the similarity method.

help for studying otherwise largely inaccessible magmatic systems. Aside the relative ease of deployment of fibre optic cables, depending on the environment, and centralized data recovery, the main advantage over conventional (sparser) arrays is the large-N sensitivity and the wave-field separation potential. We decoded the complex volcanic wavefield using the spatially dense information of DAS. New information of the complex wavefield can be decoded, enabling the applicability and future development of novel means of processing and data analysis. We are able to detect weaker signals in complex environments and we can separate different meaningful wavefield components stemming from different sources. This systematic deciphering of complex waveforms would simply not be possible with a sparse array, which would not generally detect the local response

associated with faults zones or other smaller structural features[43]. In addition, large-N nodes record ground velocity, which are by nature less sensitive to local heterogeneities, whereas strain highlight those.

Owing to the large amount of data generated, creative data management approaches[50] and big data paradigms will help boosting volcano research and monitoring with DAS. A proper understanding of volcanic activity resides in multiparametric observations; as demonstrated in this study, DAS is able to shed additional light on volcanic structure and processes. We anticipate that fibre optic technologies will become a standard for volcano research and monitoring, in particular for earthquake location, small signal detection, detailed structural imaging and a more acute understanding of the dynamics underlying magmatic processes.

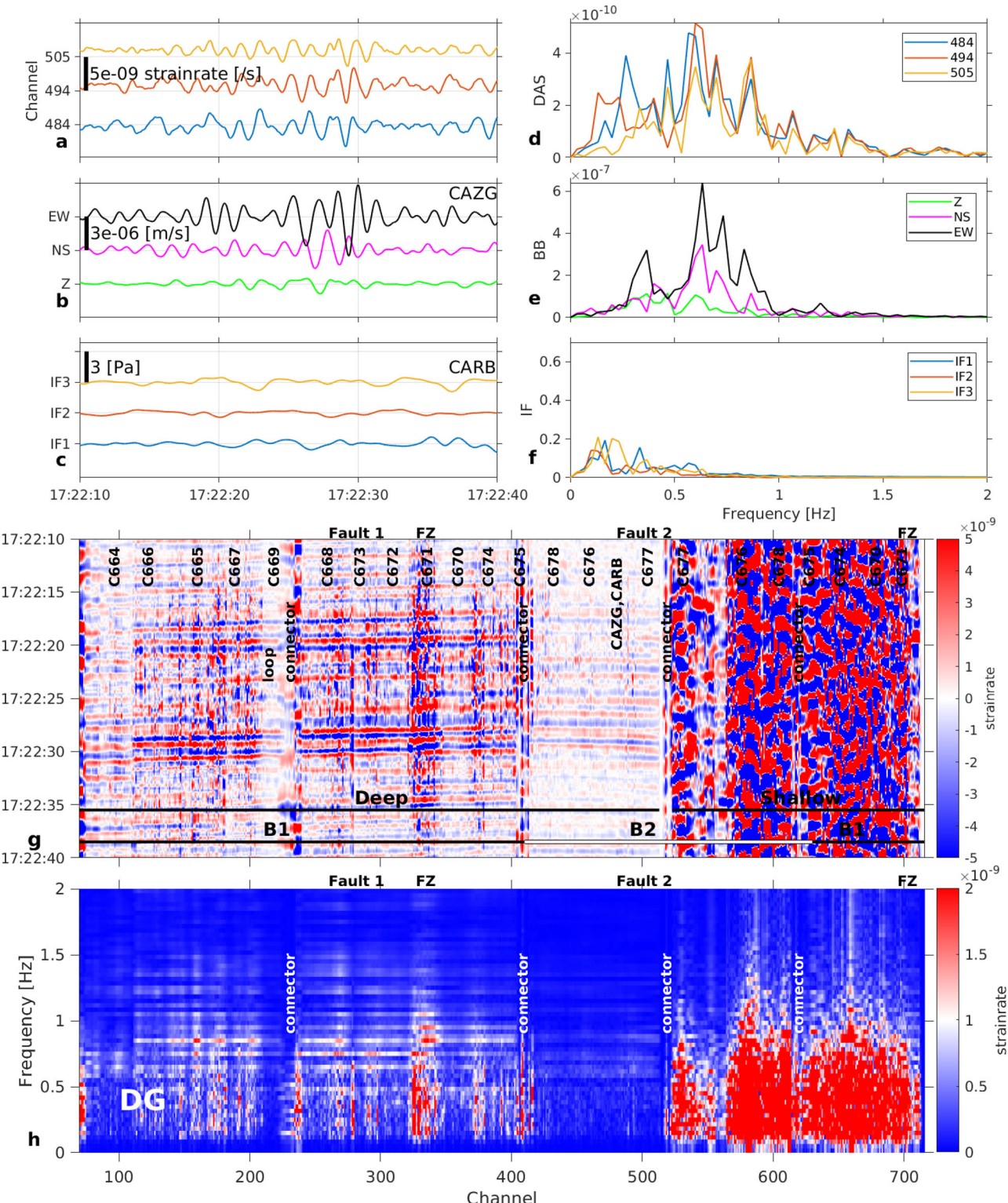

**Fig. 7 Detailed records within the tremor of a degassing (DG) event.** Records are filtered in the range 0.1–0.6 Hz. Note that DG event records have higher frequencies, which are filtered out in this figure. Unfiltered signals are shown in Supplementary Fig. 14. DG events do not exhibit any infrasound signal in our records. **a** Strain rate from distributed acoustic sensing (DAS) records at channels 484 (blue), 494 (red), and 505 (yellow), corresponding to positions of infrasound sensors in (**c**). Fibre channel position accuracy ±3 m (Method: DAS interrogator, fibre optic cable and conventional sensor network characteristics). **b** Velocity seismograms from broadband seismometer CAZG (Supplementary Table 3), near DAS channel 494. **c** Pressure records from infrasound sensors CARB-IF1, 2, 3. **d** Strain rate (a) spectra. **e** Ground velocity (b) spectra. **f** Pressure (c) spectra. **g** Strain rate record at the 710 DAS channels along the 1.3 km fibre. B1 and B2 are the two geographically distinct branches in Fig. 1. FZ: fault zone (~50 m width), at channels 315–340 (deep cable) and channels >700 (shallow cable). **h** Strain rate-frequency distribution along the cable.

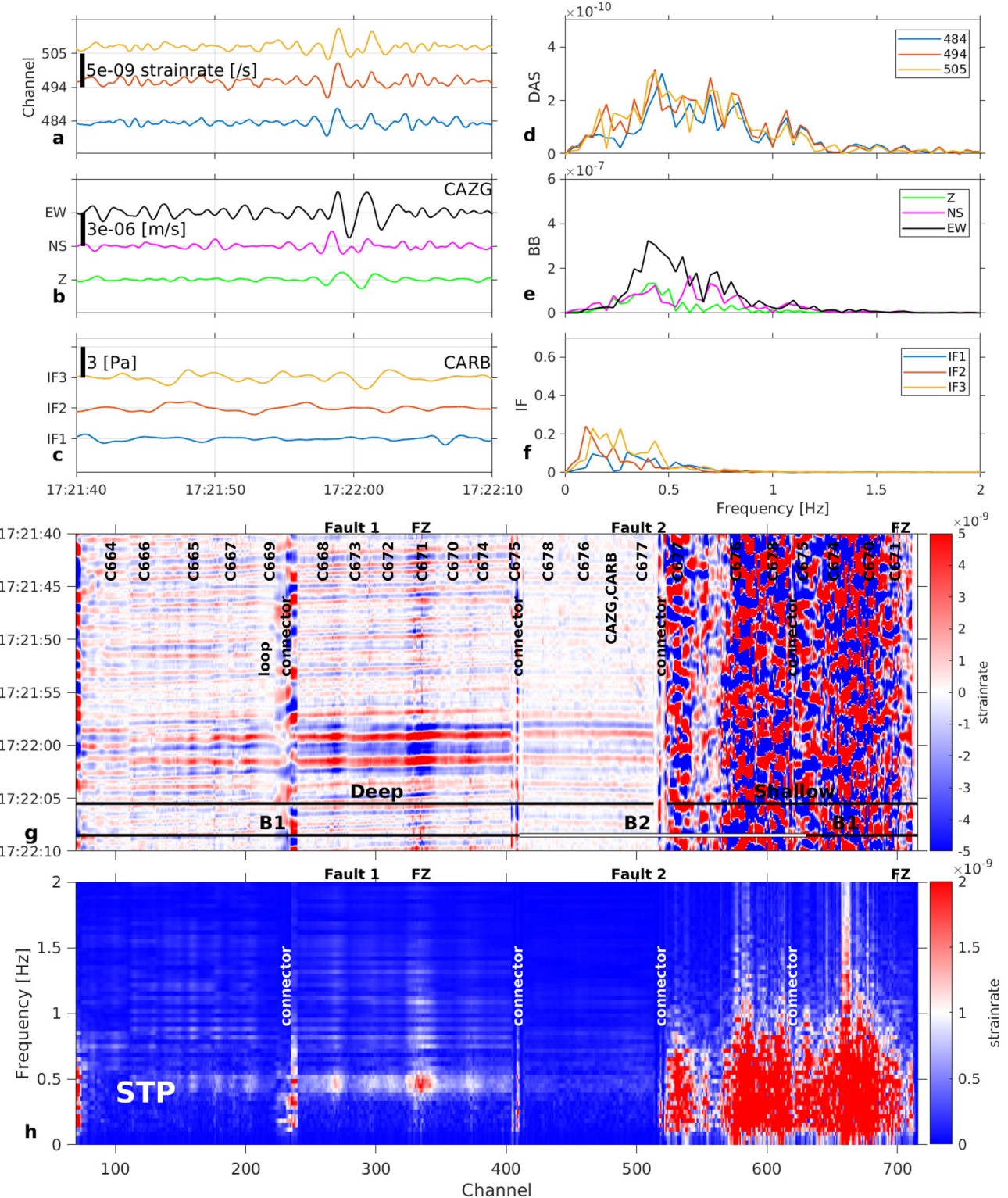

**Fig. 8 Detailed records within the tremor of a Single Tremor Pulse (STP) event.** Similar layout as in Fig. 2. Records are filtered in the range 0.1–0.6 Hz. STP events do not contain higher frequencies. Unfiltered signals are shown in Supplementary Fig. 15. STP events do not exhibit any infrasound signal in our records. **a** Strain rate from distributed acoustic sensing (DAS) records at channels 484 (blue), 494 (red) and 505 (yellow), corresponding to positions of infrasound sensors in (c). Fibre channel position accuracy ±3 m (Method: DAS interrogator, fibre optic cable and conventional sensor network characteristics). **b** Velocity seismograms from broadband seismometer CAZG (Supplementary Table 3), near DAS channel 494. **c** Pressure records from infrasound sensors CARB-IF1, 2, 3. **d** Strain rate (a) spectra. **e** Ground velocity (b) spectra. **f** Pressure (c) spectra. **g** Strain rate record at the 710 DAS channels along the 1.3 km fibre. B1 and B2 are the two geographically distinct branches in Fig. 1. FZ: fault zone (~50 m width), at channels 315–340 (deep cable) and channels >700 (shallow cable). **h** Strain rate-frequency distribution along the cable.

## Methods

**DAS, optical fibre and conventional sensors**. We designed our network to benefit from two fibre optic cable features: data transmission and DAS measurement. In order to transmit data from a 4 broadband seismometer array, we used a fibre optic cable connected to each seismometer and to the internet gateway at the Pizzi Deneri Observatory (Fig. 1). The cable consists of several 200–300 m long segments of 12 multimode fibres (OM3) connected with SC/PC connectors near the broadband seismometers. We connected an iDAS interrogator ("Ella", Serial number #14030, Silixa) to one of the free fibres in each segment to acquire strain rate data. The SC/PC connectors generate light reflections introducing spurious measurements at near channels. We used a portable power generator needing regular supervision and refuelling, and therefore acquired only during day time, except in the last 2 days when recording was more continuous (Supplementary Fig. 1a). The gauge length is 10 metres, the sampling frequency 1000 Hz, and the spatial sampling set to 2 m. In order to exploit the fibre optic sensing directionality, we buried the cable along two different directions (Branch B1 and B2, Fig. 1) with total length ~1.3 km (900 m at 15–25 cm depth and 400 m at the ground surface). This configuration allows variable sensitivity to specific volcanic activity (Supplementary Note 2, Supplementary Tables 1 and 2). We demonstrate that, even very close to the surface, the cable is well coupled (Supplementary Fig. 1, Supplementary Note 4); note that at the surface the cable is sensitive to wind (e.g., Fig. 7). DAS data represents strain rate as a function of distance to the iDAS recorder along the optical fibre. In order to reference geographically and verify observed distances from the DAS record, we performed jumps along the cable at selected places located with a portable GPS device (Supplementary Fig. 3). We assigned the geographical position of each jump (within 1 m accuracy) to the closest DAS channel. Then, we linearly interpolated the geographical positions in order to assign a georeferenced location to all channels between adjacent jumps. The final location accuracy is on the order of ±3 m. In order to validate the DAS records, in addition to the broadband seismometer array (4 Güralp CMG3-ESPC, 120 s), we temporarily deployed 15 geophones (3 components, Sensor Nederland PE-6/B, 4.5 Hz) and 1 additional broadband seismometer (Nanometrics Trillium Compact, 120 s) along the cable at ~25 cm depth (Supplementary Table 3). Gaia dataloggers digitized Güralp broadband seismometer data at 100 Hz and sent records via fibre to the observatory. Cube dataloggers digitized and recorded locally Trillium Compact and geophones data at 200 Hz. We also deployed two arrays of 3 BSU (Boise State University) infrasound sensors[35] at the surface along the cable, digitized at 200 Hz and recorded locally also with a Cube datalogger.

**DAS strain rate and strain validation**. We use three methods to obtain strain from seismic records. For all, we compare the strain (rate) as measured by DAS with independent estimates of strain (rate) based on the geophones and broadband seismometer records. Both seismic data (after instrumental response is corrected for) and DAS data are bandpass filtered (0.1–15 Hz). DAS strain rate records are then integrated to strain by simple numerical integration in time (assuming the initial strain is zero). In the first method (Phase velocity), velocity data from each seismometer is converted to strain by using the time series at a single location[51]. Under the plane wave approximation, the particle velocity is related to strain as $\varepsilon_x = -p\dot{u}_x$, where $\varepsilon_x$ is the strain, $\dot{u}_x$ the particle velocity projected along the cable direction $\vec{x}$ and $p$ the apparent slowness. We compare strain derived from broadband (Supplementary Fig. 4a) and geophone (Supplementary Fig. 4b) velocity data with the DAS strain measurements at the co-located channels. We select the nearby channel with the highest cross correlation. This method requires a local estimate of the phase velocity[52]. The apparent wave propagation velocity $c_x = \frac{1}{p}$ is determined by the best-fit between strain $\varepsilon_x$ from the integrated DAS strain rate records and particle velocity $\dot{u}_x$ from the seismometer records. Along the cable path, $c_x$ varies from 400 to 1100 ms$^{-1}$, which are upper limit values of the true propagation velocity. It is worth noting lower values correspond to the fault zone close to geophone C671 (Channel 333, Supplementary Table 3). In the second method (Spatial displacement gradient), we take advantage of the dense array deployment of geophones: we derive the strain $\varepsilon_x$ along the cable direction, in the small strain limit[53]. We calculate the scalar-product (denoted·) between the cable direction $dx$ and the difference $du$ of the two displacement vectors, i.e., $\varepsilon_x = dx \cdot du$. This method allows for a direct comparison between strain derived from seismic data and strain derived from DAS data without requiring the ground phase velocity[54]. We compare the strain derived from geophones with

(i) the stacked DAS strain signal from channels between the 2 geophones,
(ii) the DAS strain signal sampled at the midpoint channel,
(iii) the DAS strain signal sampled at the channel with the highest cross-correlation with the strain derived from the two geophones.

The minimum normalized RMS errors yield a better agreement selecting the DAS signal with the highest cross-correlation (Supplementary Fig. 4c). The discrepancies between estimates and exact values depends[55] on the ratio between station distance $d$ and signal wavelength $\lambda$ as $\sin(\frac{\pi d}{\lambda})/(\frac{\pi d}{\lambda})$. Since the average geophone distance is ~50 m in our field deployment, signal wavelengths larger than 100 m could be approximated by this method with an error of less than 35%. Based on the range of phase velocities calculated earlier, this approximately corresponds to frequencies of 4–10 Hz. Those comparisons can be equally applied to the strain rate (velocity) as it is simply the time derivative of the strain (displacement). In the

third method (Strain rate over gauge length), DAS strain rate is defined at the midpoint of a gauge length L as the difference between velocities measured at the extremes -L/2 and L/2 and divided by L. For a line segment of length nL the summation of the strain rate over the n points within the segment can be written[56] as

$$\dot{\varepsilon}_x\left[-\frac{(n-1)L}{2}\right] + \cdots + \dot{\varepsilon}_x[0] + \cdots + \dot{\varepsilon}_x\left[\frac{(n+1)L}{2}\right] = \frac{\dot{u}\left(\frac{nL}{2}\right) - \dot{u}\left(-\frac{nL}{2}\right)}{L} \quad (2)$$

The DAS device is configured to measure strain rate with a gauge length of 10 m and a spatial sampling of 2 m. Therefore, there is an overlap of strain probed for successive DAS traces. In order to independently sample strain rate in space, we resample the DAS strain rate traces every gauge length (5-channels) over the cable segment between two consecutive geophones. Supplementary Fig. 4d shows the comparison between the left term (derived from DAS) and the right term (derived from the geophone) of Eq. (2). The three methods generally yield a good agreement between the direct DAS strain measurements and the strain estimates from seismic sensors. The average amplitudes fit well, although discrepancies are observed at higher frequencies, i.e., shorter wavelengths approaching the gauge length.

**3D wave propagation modelling with topography in complex media**. Adapting a finite difference code[57], we simulate wave propagation in 3D complex media (viscoelastic, including topography) to illustrate the main features of the seismic wave and strain field (polarization and amplitude variability) recorded along branches B1 and B2 at Piano delle Concazze. We quantify the seismic wave and strain field distortion caused by 3 main features: topography[58], a fault zone and 3D tomographic models for P- and S-wave velocities and P-wave for attenuation[34] superimposed by a shallow low velocity layer following the topography. Our background model uses standard values for Etna[59–61]: P-wave velocity $V_P = 3500$ m s$^{-1}$, $V_P/V_S = 1.7$, quality factors $Q_P = 100$ and $Q_S = 80$, to account for attenuation of P- and S-waves, respectively, and density d = 2670 kg/m$^3$. The fault zone with lower density (600 kg/m$^3$), and lower velocities ($V_P = 1100$ m s$^{-1}$ and $V_P/V_S \sim 1.8$) is introduced in some of the models. The low velocity layer is 100 m thick with $V_P = 1700$ ms$^{-1}$, $V_S = 1000$ ms$^{-1}$ and $Q_P = 75$. All the above values are approximate and here, the aim is only to explain first-order features of wave-field distortion and local strain (rate) amplification under simple controlled conditions. The computational domain uses a 4th order staggered grid scheme with a spatial discretization of 30 m in both EW and NS directions and 15 m vertically. We used a 2D Cardinal B-splines to adapt the Etna topography obtained from the Pleiades satellites[58] and a trilinear interpolation method to embed the tomographic models[34] into the computational grids. An explosive source is modelled as a single vertical force located at NSEC. Indeed, a downward force is required to compensate for the upward momentum of the volcanic ejecta, justifying the use of a single vertical force. We compare observed data (geophone velocity and fibre strain rate) and results of the simulated seismic wave propagation (waveforms, particle motion and strain-rate) for 8 models (Supplementary Figs. 5, 6 and 7). The 8 models cover the systematic investigation of 3 main features. Each combination is considered to evaluate qualitatively what is the relative contribution of each feature. We adapted the code[57] to compute the strain rate tensor from the spatial derivative of the velocity output. As the density of the fibre channels (every 2 m) is larger than the computational grid node density (every 30 m), we linearly interpolated the strain rate tensor components at the cable locations using computed strain rate values at the grid nodes. We then projected the strain rate tensor along the cable at each channel location after smoothing between grid nodes. Although the real surface velocities may be even slower than those in the tested models, our results illustrate that the tomographic models with shallow velocity layer has the largest influence in order to approach observed relative amplitudes strain rate (models e to h in Supplementary Figs. 5, 6 and 7) between branch B1 and B2. Similar to the observations, we obtain larger amplitudes in strain rate for models incorporating the fault zone (models c, d and g and h in Supplementary Fig 7). From the infrasound record, we derived the pressure amplitude at the first nodes of the grid and apply a single force. A vertical force strength of ~1 to 2 × 10$^9$ N (0.4 to 0.8 MPa at the first computational grid nodes) leads to a good agreement between synthetic and observed DAS amplitudes at the cable. An isotropic source (explosion) results in similar waveforms but synthetic amplitudes do not match the observed amplitudes. Observations last longer than the synthetics resulting from the 1.5 s Ricker source, suggesting that the source time function may be more complex and that additional structural features not considered in our modelling distort the wavefield.

**Ground velocity estimation**. We used two methods. The first method (Inter-channel travel-time) approximates the propagating seismic wavefield as a plane wave. The inter-channel travel time difference is $\tau_S = \frac{d_S}{V_h}\cos\gamma$, where $d_S$ is the inter-channel distance (defining a fibre segment S), $\gamma$ the angle between the cable and the seismic wave directions and $V_h$ the apparent velocity. This relationship holds for any S and hence can be applied to estimate the velocity distribution at high spatial resolution along the fibre. It is not possible to separate the subsurface velocity from $\gamma$. However, if we assume that $\gamma$ is constant along linear sections of the cable (e.g., B1 or B2), variations of $\tau_S$ are only related to medium velocity variations. For each segment along the cable, we obtain $\tau_S$ by computing the maximum of the normalized cross-correlation between seismic signals taken for various inter-channel

distances during the explosion event (Supplementary Fig. 10). The $\frac{d_s}{\tau_s} = \frac{V_h}{\cos\gamma}$ values show a high variability ranging from 330 to 1100 m s$^{-1}$, in agreement with the values of apparent velocity estimated in Method: DAS strain rate and strain validation, phase velocity. The derived velocity estimates represent projected quantities that depend on $\gamma$ and near-surface media properties and the cable geometry. Thus, they should be viewed as upper limits for the dominant seismic mode (Rayleigh waves). Lower values are observed within fault zones.

The second method (multichannel analysis of surface waves, MASW) provides dispersion curves, which can be inverted to obtain vertical 1D shear wave velocity profiles[62,63]. For each jump performed along the cable, we define a forward sub-dataset (records toward increasing channel numbers) and a backward sub-dataset (records toward decreasing channel numbers). For consistency between the different jumps, we limit our analysis to only 50 channels (100 m) of 1000 samples (1 s), although propagation of dispersive modes for some jumps could be observed on up to 100 channels (200 m). We compute the dispersion curve using a phase-shift method[64], pick the observed modes in the dispersion curves and invert them simultaneously to derive a vertical profile of shear wave velocities for the forward and the backward subsets. We perform a Markov chain Monte Carlo inversion[65] in order to sample the posteriori probability function for interface depths of the layers and their shear-wave velocity (Supplementary Fig. 11). To gain stability, we used an enhanced version of the records (Method: coherent field separation and data enhancement).

**Coherent wavefield separation and data enhancement**. Summation-based coherence analysis is known to be noise-robust and physically justified, as long as sufficiently dense spatial data sampling can be ensured[66]. By means of the semblance norm[67], we estimated the local coherence of the recorded strain rate. Specifically, we evaluated semblance $S$ on a fine predefined slowness grid (individual values indicated by index $k$ with $p_k \in [-0.008; 0.008]$ s/m and increment $\Delta p = 0.0002$ s/m) for each fibre position $x_0$ and recording time $t_0$

$$S_k(x_0, t_0) = \frac{1}{N} \frac{\sum_{\delta t} \left[\sum_{i=1}^{N} D(x_0 + \Delta x_i, t_0 + p_k \Delta x_i)\right]^2}{\sum_{\delta t} \sum_{i=1}^{N} D^2(x_0 + \Delta x_i, t_0 + p_k \Delta x_i)} \quad (3)$$

where $D$ is the DAS data amplitude, $\delta t = 0.05$s is a centred time window, $i$ represents the channel index and $N$ denotes the number of channels spanned by the local aperture with relative fibre distances $\Delta x_i \in [-25; 25]$m. Utilizing the semblance value, which acts as a data-derived measure of trust (ranging from 0 for incoherent energy to 1 for perfect data coherence), we derive the coherent reconstruction amplitude $\underline{D}$ by weighting and integrating the individual contributions of all slowness values[68]

$$\underline{D}(x_0, t_0) = \sum_k S_k(x_0, t_0) \sum_i D(x_0 + \Delta x_i, t_0 + p_k \Delta x_i) \quad (4)$$

By repeatedly substituting $D$ with $\underline{D}$, summation over the full considered slowness range highlights lower frequencies, leading to a natural hierarchical separation of the long-period volcanic signal from the DAS record (Fig. 4a). We constrained the path summation to the positive slowness branch, i.e. to values ranging from 0.0002 s/m to 0.008 s/m to only reconstruct the transmission component of the infrasound-induced signal, whose adaptive subtraction from the enhanced full wavefield then gives the weak separated reflection propagating in the opposite direction (Fig. 4b). To stabilize subsequent processing, the jumps investigated in Supplementary Fig. 11 were likewise separated and enhanced following the described procedure. In contrast to conventional f-k filtering, coherence analysis aids noise suppression and data regularity and can be more flexibly tailored towards specific wavefield components e.g., by also incorporating wave front curvature information[30].

**Detection methods for continuous monitoring**. We applied three detection methods to our DAS records: the average STA-LTA function, absolute amplitude stacking and local similarity (Figs. 5 and 6). They all are based on estimating a specific characteristic function from the multichannel records. The event detection is then performed by computing the median absolute deviation (MAD) of each characteristic function, and an event is declared when a threshold defined as the median plus 3 times the MAD over a 5-minute sliding time window is exceeded. In the first method (Average STA/LTA function), the ratio between Short-Term Average (STA = 0.7 s) and Long-Term Average (LTA = 10 s) is computed for each channel along the fibre, and then averaged. In the second method (Amplitude stacking), we take advantage of the high information density provided by DAS. For each time sample of the DAS records, we perform summation of the absolute amplitude values over all the channels. The third method[69] (Local similarity), is based on local similarity among signals from nearest stations. The wavefield associated to a common source is expected to be similar for spatially close stations, unlike random noise fluctuations. Therefore, owing to the dense spatial sampling, this method is ideally suited for continuous DAS data analysis, as it allows to efficiently discriminate between coherent signal and noise. Depending on the signal frequency, DAS channels which are too close (2 m) are not independent due to the 10 m gauge length. Waveform similarity is then always high, and thus not useful for discriminating coherent signal from noise. As the cable is more sensitive in the fibre axial direction, the similarity will be higher for signals from channels belonging to linear cable sections. We thus quantify the local similarity separately on branch B1 and B2, by computing the average normalized cross-correlation

maxima of data among channels 60 m apart over a 10 s window sliding with time shifts of 0.5 s. Then, the characteristic function representative of the entire network is obtained by stacking the local similarity values. For all three methods, we tested several frequency bands. Good detection performance is obtained for 0.1–5 Hz for STA-LTA and absolute amplitude stacking and 0.1–0.6 Hz for local similarity. The frequency content of the volcanic tremor (0.5–5 Hz), which is quasi permanent introduce a high similarity between close traces, and therefore the detection of distinguishable triggers is prevented. The most discriminative frequency band was found to be 0.1–0.6 Hz, which removes the main frequencies of the persistent tremor, and keeps only tremor bursts of larger amplitude and lower frequency.

## Data availability

The strain rate, infrasound and seismological dataset[70] generated and analysed during the current study have been deposited in the Geofon database under accession code 9N (https://geofon.gfz-potsdam.de/waveform/archive/network.php?ncode=9N&year=2018).

## Code availability

Codes are available upon request to the authors.

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

## Acknowledgements

We gratefully acknowledge help of the following people and institutions; David Bruhn supported the initial input; Michael Weber and Christian Haberland for fruitful discussions; Christian Cunow for the help with the laboratory measurements. INGV staff composed of S. Consoli, D. Contrafatto, G. Larocca, A. Messina, D. Pellegrino, M. Pulvirenti, S. Rapisarda for great help with geophone, broad-band sensors deployment, and hard work for cable deployment; The Society Guide Vulcanologiche Etna Nord for allowing us to use one of the videos from the summit activity and for collecting scoria samples; Parco dell'Etna and the municipalities (Linguaglossa and Castiglione di Sicilia) for authorizations to deploy the cable and instruments at Piano delle Concazze. Geophones, broad-band seismometers and data logger equipment are from the Geophysical Instrumental Pool of Potsdam (GIPP). The iDAS interrogator "Ella" was lent by Silixa Ltd. Geofon team (Susanne Hemmleb and Javier Quinteros) helped a lot in building the data base repository. This work received funding from the GeoForschungZentrum Potsdam (projects DAS@SEA and InfraDAS) and INGV.

## Author contributions

P.J., G.C. and L.Z. designed, planned the experiment and performed the field measurements. P.J. guided the whole experiment, wrote the first manuscript draft and the final version of the manuscript. P.J. and G.C. analysed the data and produced most results shown in the manuscript and built the database. B.S. produced the wave-field separation results and wrote that part of the manuscript. T.R. coordinated the laboratory analysis and supported during the evaluation of the cable design and coupling issues. A.C. supported remotely DAS operation with "Ella" and organisational aspects. F.T. suggested aspects of the deployment strategy and

contributed to the interpretation. C.K. and E.P. supported the initial idea and gave strong input in the interpretation and future perspectives.

## Funding

## Competing interests
The authors declare no competing interests.
