## [Peer Review File · Nature Communications]

Fibre optic distributed acoustic sensing of volcanic eventsREVIEWER COMMENTS

Reviewer #1 (Remarks to the Author):

The submission by Jousset et al. documents a seismology and infrasound study of Mount Etna, Italy, using fiber optic cables and distributed acoustic sensing (DAS). For comparison, traditional seismological and infrasound data are acquired and analyzed, as well. The study demonstrates the potential of DAS data for detecting and locating seismic events related to volcanic activity. The authors focus on high-quality DAS data related to volcanic explosions and make a convincing case of nonlinear coupling of infrasound waves to seismic waves. Moreover, they highlight complicated wave propagation controlled by the volcano's shallow seismic velocity structure, topography and 3D effects such as fault interfaces.

I find the analysis of the DAS data very thorough going beyond simple documentation of observations. My main criticism is that the reader (in particular a non-volcanologist like myself) is likely left wondering how typical the presented findings may be for volcanos in general. This could be brought out better in order to underline that this study is a milestone for volcano-related research. This applies in particular to the final part of the paper, where the authors focus on detectability of two event types using DAS. I suggest starting this part with a discussion of the meaning of these events and the value of being able to monitor them. Then the detection approaches will be more appreciated.

In its current version, the manuscript emphasizes that the described DAS approach is valuable in the sense of allowing remote monitoring, which does not put researchers at risk during the cable installation. I do not follow this argument, because (1) the cable was placed closely to the crater and (2) seismometers and infrasound sensors yielded the complementary measurements to the DAS approach and were collocated with the cable. So it is not clear why DAS has an advantage over conventional sensors in terms of safety. That said, the study is interesting and sound such that the safety argument is not needed anyway. It will be clearly of interest to a wider audience as it demonstrates the potential of DAS data for volcano monitoring.

The manuscript is clearly written, only some English editing is necessary. Besides the points above I have few points of criticism and mostly minor remarks. Although addressing my review may require some text modifications and/or additional explanations, I find this submission is close to publication quality.

Fabian Walter.

GENERAL REMARKS

As mentioned above, the authors should better embed their findings within the existing volcano literature, especially that pertaining to Etna volcano. What is the role of the explosion events and the two event types described towards the end of the paper? What are the source mechanisms of these signals? What is known about the layer of scoriae? Is this a failure-prone layer that plays a role in lahar or other types of dangerous mass movements? What is the benefit of knowing the layer thickness distribution? How are tremor pulse events and degassing events generated and why are they relevant? The sentence on Lines 264-265 seems rather weak and general and should be solidified with discussion on volcano

seismology. Similarly, what is the physical meaning or interpretation of the specific detection statistics in Figure 4c? For the broad audience of Nature Communications, additional explanations of these events are necessary. Have these events been studied on Etna and elsewhere previously? Also, why were the explosions, which apparently leave a clear seismic signature on the DAS record not considered as a third event type for automatic detection?

SPECIFIC REMARKS

Abstract: infrasound is not mentioned although it plays a central role in the study.

Line 45 and elsewhere: What is meant by “disruptive progress”?

Line 51-52: How do infrasound waves reveal source depths?

Lines 109-112: I do not understand where the maximum propagation distance of 300 m comes from.

Paragraph starting on Line 114: If I understand correctly the authors refer to wave polarization relative to the cable orientation when arguing that segment B2 should record higher amplitudes than segment B1. On the other hand, B1 is closer to the source. Is this relevant?

Discussion of Figure 2: It may be worth pointing out that the infrasound frequencies shown in Figure 2a do not show up in the cable (Figure 2h). So the infrasound signal in the DAS record is really an induced signature rather than a result of cable-air coupling.

Line 148: I would refrain from the word “perfect” for any type of measurement-based evidence.

Lines 156-157: The higher frequencies are suppressed rather than absent.

Line 16: “material of 1 to 3 cm diameter” \diamond “grains of 1 to 3 cm diameter”

Paragraph starting on 165: The shift in resonance frequency peak is the most prominent feature of Figure 2. The derived thickness and variations thereof should be compared to previous studies or alternative measurements (e.g. H/V ratios or autocorrelations of seismic point measurements).

Paragraph starting on Line 202: For the location and the azimuth determination of the fault some more evidence should be given, ideally with a supplementary figure.

Paragraph starting on Line 293: This contains absolute statements such as “first time” and “unprecedented accuracy”. To my mind this deserves justification in terms of 1-2 discussion sentences.

Line 312: “Weak” is a weak qualification. With respect to what?

Line 314: which conduit?

Line 315: "information density" is unclear.

Lines 318-320: The DAS cable has to be installed, too.

Lines 322-325: I find these statements rather generic for a conclusion. Underground deployment of a cable seems laborious compared to geophone installations. The big data paradigm is generally applicable to DAS and many modern seismology applications. I would stay a bit closer to the findings of the present study and open up the perspective from there.

Line 533: Throughout the text I was wondering if the cable was there first to transmit seismometer data or if seismometers were used to magnify the cable installation. This would be a good location in the text to clarify this.

Line 557-559: Can the loop effect be highlighted better?

Line 593: Where does the bandwidth for the propagation velocity come from?

Line 601: Which "two displacement vectors"?

Line 603: Rewrite "avoiding the knowledge".

Line 610: "better agreement" should ideally be quantified. Also, the following paragraph requires a reference upon which the numbers and mathematical expression are based.

Method 3: It would help the reader to be reminded that you are referring to seismic waves transmitted into the ground by infrasound signals.

Line 656: Why is the explosion source modeled as a single force (rather than an isotropic moment tensor)?

Lines 704-795: rewrite "records being toward ..."

Line 709: Why "successively"?

Line 737: Equation (3): why are there such large spaces in the equation?

Line 758: overpassed \diamond exceeded

Lines 782-783: "most meaningful for set of channels ..." not clear what this means.

Lines 788ff: The permanent volcanic tremor is an entirely new concept. Is this another source type? Also, the sentence starting with "The permanent ..." should be rewritten.

Lines 987-988: I would avoid accumulation of more than three nouns.

Line 1022: "spectrogram with time": unclear

Line 1059: stroke \diamond struck

Lines 1065-1066: "systematic differences in the amplitude with depth": unclear

Lines 1078ff: how were shear forces and yield strength calculated/determined?

Lines 1088ff: What does reorganization mean? Sliding of grain boundaries? Also, "are likely to provide sufficient" sounds clumsy.

Line 1107: "was not be recorded": rewrite

Lines 1120-1121: permanent frequency \diamond sustained resonance

Line 1128: "following the deep cable backward" is unclear

Lines 1132ff: "seems rather very long period" sounds clumsy. Same for "matches approximately well".

In several parts of the text the authors use the term "elongate". Is this used synonymously for "stretch" or "strain"? I would keep the terminology as limited as possible to avoid confusion.

The piano tests receive a lot of attention. Why is the video necessary? Tuning a piano should be a familiar concept to all readers requiring little explanation. The supplemental figure suffices to my mind.

Video "Explosion at NSEC on 5th September 2018 at 10:54" cannot be opened.

FIGURES

Figure 1: A scale bar for the bottom right inset is needed. What is the yellow box in this inset?

Figure 3: It would help to see another box in the (a) panels showing the zoom-ins of panel (b).

Figure 4: The y-axis label should be defined in the caption.

Extended Data Figure 3: Define y-label in upper panel.

Extended Data Figure 4: Upper panels: The "(2 m)" in the x-labels is confusing. It sounds like 2-m-high jumps. "MASW" should be written out once in the caption (same for acronyms in other figure and table captions).

Extended Data Figure 6: I could not verify the last two sentences in the caption and am not sure I understood the content and how to compare model and data.

Supplemental Figure 1: I would label the notes in the upper panel.

Supplemental Figure 3: I cannot verify the resonance at 25-30 Hz. Rather, a concentration of spectral energy?

Reviewer #2 (Remarks to the Author):

Review to the manuscript: "Fibre optic distributed acoustic sensing of volcanic events" by Jousset et al.

The authors deployed a Distributed Acoustic Sensing (DAS) on Mount Etna, Italy and demonstrate the potential of this "new" technique to detect and analyse volcanic events as well as imaging volcanic structural features. The authors compare the results with a set of broadband, geophone and infrasound sensors. They provide videos, which made the manuscript pleasant to discover. The DAS is applied for the first time on a volcano, for the positive data resulting from the survey and the promising future applications of this technology, I recommend the manuscript for publication after taking into considerations the following comments.

The comments:

Lines 121-123: Using 3D wave propagation simulations, we demonstrate that Etna topography and local volcanic structures explain the wavefield variability of both particle motions (Extended Data Fig. 5) and strain rate amplitudes (Extended Data Fig. 6). It seems from these two reported figures that you show the simulated results from modelling, but do not present a proper comparison, where I would expect differences between observed and simulated data as well as the computation of a misfit measure. For Ext. Data figure 5, enhance in the caption or in the graph itself that points a-e are for synthetic results and f is for observed data. For Ext. Data figure 6, observed data should be added aside or below to better see how the data can be compared.

Line 129- Fig2: This is not easy to understand on Fig 2g and h where the channels are located, in reference to Fig 1. Why is B2 inserted into the profile B1?

Lines 228-243 Fig. 3: As you also mentioned in the caption, the theoretical strain rate arrival times are barely visible on the subplot c. This could be brought out with different colors and with an onset on a particular time interval.

Line 241: Is the explosion seen in supplementary video 2 associated with the records in this Figure 3? If so, the time occurrence is not consistent 10:54 for the signals and 11:00:42 for the small degassing. Would it be possible to add a time scale on the video and/or the signals as in supplementary video 1?

Line 252: Extended Data Fig 7: could you add the signature of the strain rate for the two types of signals in this figure or in the manuscript? (As in Fig 2a)

Lines 260-270 : statistical analysis: Was it performed without distinguishing the two types of events (TPE and DGE)? Figure 4b and c show similar information, Figure b could be moved to supplementary material. What is the importance in the manuscript to show the results from the 3 types of detection method?

Was also the amplitude of the TPE and DGE investigated?

What about the location of those signals? Are they all coming from the same source? Does it show activity in the different craters or localized to one in particular for the investigated time interval?

291 Advantages/ disadvantages of the DAS should be discussed more in details in this conclusive paragraph : limitations/installation set-up/ what could still be improved for further

studies?

Supplementary material: The structure of the supplementary material (methods, notes and extended data) should be improved. This would be easier for the reader to have, in the manuscript, the reference to one supplementary note/file/figure and not several as for the example: line 158 (supplementary Fig. 1, Supplementary note 2 and supplementary video 3). At least supplementary information should be gathered. One could in the manuscript refer to the supplementary notes, that in turn refer to the associated figures.

Also, attention should be taken for introducing the note/figure/table references in the order of appearance:

ex line 106: supplementary Table 3 is mentioned before supplementary table 2

ex lines 123-124: extended data Fig. 5 and 6 before extended data figures 2-3-4.

As it is, it makes the reading a bit confusing.

Reviewer #3 (Remarks to the Author):

Review of manuscript "Fibre optic distributed acoustic sensing of volcanic events" by Jousset et al. for publication in Nature Communications

The manuscript shows an initial application of Distributed Acoustic Sensing (DAS) to volcanic environments using data from a cable deployed within a few km distance from the summit crater of Mt. Etna. Results are presented for (i) subsurface imaging using resonances caused by volcanic explosions (this though turns out to be simply estimating the scoria layer thickness, which is not very exciting), (ii) wavefield recordings of seismic and infrasound signals and their separation, and (iii) detection of volcanic signals (monitoring). The goal is to show "DAS as a new tool for volcanology", the title of the discussion section.

DAS is a new technology that is starting to be used in a wide array of applications. The optical cables measure strain rate and the cable acts as a dense linear network of strain sensors. That is really cool and offers opportunities for wavefield monitoring and the detection of small signals (through stacking or other signal enhancement strategies) previously not (or basically not) available. DAS will play a significant role in geophysics (seismics, seismology) and volcanology. The lead author (and some of his co-authors) have published a similar, initial study introducing DAS as a tool for imaging of seismological and structural features (Jousset et al. 2018, Nature Comm. 9, article 2509) and this manuscript is trying to do the same for volcanology.

The example applications show the usefulness of DAS. No question. What they do not show is was DAS necessary to obtain them or did DAS make it easier to reach the results or are they better constrained? It seems, the same data analyses could have been performed with seismometers and/or infrasound sensors (networks, small arrays) at the same 'safe' distance from the craters. The manuscript repeatedly mentions that DAS may make studying a volcano a safer undertaking but I do not see why (lack of a compelling argument). I did find the use of an obituary, to say the least, a little disturbing when talking about safety and I would hope the authors could come up with a better argument. DAS will not replace or aid geochemistry (and gas measurements).

The manuscript is well written. The sections Extended Data and Supplementary Information

contain a few more grammar and word choice issues than the rest of the manuscript and should be looked at again; sorry I'm not providing a list (L758 'overpassed' do you mean s.th. like 'exceeded'? L941 'profile C672bwd', should this be 'profile C672bkw'? There are a few more small things like that)

The authors used an impressive range of analyses methods and I have no significant issues with the methods.

(i) I found the wavefield separation example particularly interesting. I'm fairly sure though, that infrasound data by themselves suffice to determine the source crater for the explosions?

(ii) The argument about the need to use Etna topography and possibly a fault zone and low near surface velocity is not particularly convincing (Extended Data Fig. 5) and I wonder whether showing particle motions (of what exactly? For Flat Topography is that simply P and S or also surface waves? Would be good to explain what's shown for the simplest case); most obviously the horizontal component motions do not work (flat earth synthetics vs. observed) but none of the more complex models really does a good job either.

(iii) The DAS monitoring part detected two types of recurring signals that were detected with STA-LTA and stacking, and with 'local similarity', respectively; it is not clear what these signals are and where they come from; the authors seem to postulate one is a (deeper) tremor pulse event (lower frequency) and the other a shallow degassing event (with higher frequency signal) attributed to 'a ... expression of fluid movement'. A similar temporal behavior of these signals and of LP events makes the authors suggest the two event types and LP events might be 'somehow related.' First, was DAS necessary to find these two types of signal? Second, I don't follow why similar temporal behavior statistics would lead to a relation between these signals and LP events. Third, 'whatever the interpretation ...' simply shows that DAS is not a panacea, somebody still has to come up with a viable interpretation for 'whatever' signals are being recorded.

— I do understand that it is difficult to show 'DAS works', which requires to compare with other data that record the same signal opening up critique along the lines of 'DAS is not needed.' I think what would help (me) would be a clear example that DAS shows 'more' (and smaller) signals of the same type as seen with other methods and/or that DAS recordings have a higher resolution than other methods (or something similar). The added gain from DAS currently does not come through clearly.

Operating a fibre optic cable seems to require substantial infrastructure such that the authors conclude that a cable might be connected during a crisis rather than being operated as a monitoring tool. The huge amount of data (storage and processing) currently makes it not feasible to operate DAS continuously over an extended period of time. These issues will be solved in the future, but for the present and near future fibre optic cables will be a research tool. And a research tool available only at a very few volcanoes. The authors could have made this more clear, hinted at future 'monitoring' applications and given a few specific research examples to be addressed with DAS?

L24. 'Disruptive progress ...' - This is correct but many readers might be confused by the word 'disruptive' rather than 'innovative' or 'groundbreaking' or similar.

L47. '... have been efficient to ...' - what is meant by 'efficient'?

L42-59. The paragraph suggests that DAS recordings might lead volcanologists 'to

accurately anticipate subtle, yet decisive fundamental volcanic processes'? That might be a tall order. DAS, more likely (?), will become at some, well-funded volcano observatories another tool to probe a volcano, and different datasets (incl. DAS) combined might get us 'to accurately anticipate subtle, yet decisive fundamental volcanic processes'? I would appreciate if the authors would tone down their statement.

Not sure I understand what the red bars in Supplemental Fig. 6 show. Times when the cable was 'on' and connected to the interrogator, I presume?

Point to point answer to the 3 reviewers

16.08.2021

Note: in black: reviewer questions.

In blue: our answers

Reviewer #1 (Remarks to the Author):

The submission by Jousset et al. documents a seismology and infrasound study of Mount Etna, Italy, using fiber optic cables and distributed acoustic sensing (DAS). For comparison, traditional seismological and infrasound data are acquired and analyzed, as well. The study demonstrates the potential of DAS data for detecting and locating seismic events related to volcanic activity. The authors focus on high-quality DAS data related to volcanic explosions and make a convincing case of nonlinear coupling of infrasound waves to seismic waves. Moreover, they highlight complicated wave propagation controlled by the volcano's shallow seismic velocity structure, topography and 3D effects such as fault interfaces. I find the analysis of the DAS data very thorough going beyond simple documentation of observations.

Thank you for this global positive evaluation of our work, and we are happy that the case of non-linearity is convincing.

My main criticism is that the reader (in particular a non-volcanologist like myself) is likely left wondering how typical the presented findings may be for volcanos in general. This could be brought out better in order to underline that this study is a milestone for volcano-related research.

We fully agree that the initial manuscript lacked suggestions that results from this work can be applied to any volcano and beyond. Below a list of arguments, that we included in a more concise version in the manuscript (indicated for each line number in the new manuscript):

1. Explosions and ash deposits are a common on volcanoes and elsewhere; therefore, we may expect this non-linearity excitation to happen at all volcanoes with strong explosions. Implications include triggering of events that may be sensitive to various frequencies (new lines 422-426).
2. The coupling of the infrasound signal with the ground is a topic that may interest a wide range of cases, where explosion occur at the surface, potentially also to study underground (lines 426-429).
3. The detection of small events is a general challenge on volcanoes, but also for geothermal systems, oil and gas reservoir management, making our techniques of detection quite general. (not reported in the new manuscript).
4. The use of already existing infrastructure is particularly appealing and could help reach acceptance and full appreciation of the actual cost-effectiveness / affordability of DAS (lines 440-445).
5. New types of processing / analysis become possible. Aside from cost-effectiveness, relative ease of deployment and data recovery, the main advantage over conventional (sparser) arrays is the large-N sensitivity and the separation potential - not only in a technical, but also in an applied context. We should be able to detect weaker signals and (in complex environments this might be even more important) we can separate different meaningful wavefield components stemming from different sources, which would otherwise be "mushed up". This systematic

"deciphering" of complex waveforms would simply not be possible with a sparse array (line 456-461).

6. The deployment of a fibre-optic cable can aid continuous data transmission and maintenance, and therefore also integrates well with more conventional monitoring strategies. and it may be argued that we also provide a proof of concept for what dark-fibre based monitoring in towns/cities at volcano slopes could possibly achieve (Partially reported in lines 439-444).

We have extended the discussion to address those aspects and suggest possible implications of our study and let the reader conclude that our study is "a milestone for volcano-related research".

This applies in particular to the final part of the paper, where the authors focus on detectability of two event types using DAS. I suggest starting this part with a discussion of the meaning of these events and the value of being able to monitor them. Then the detection approaches will be more appreciated.

We have introduced a description (new lines 60-97) on what event types are typically recorded on volcanoes (Volcano-tectonic earthquakes, Long-Period, tremor and explosions). Benefiting from the high spatial density of the DAS and implementing different detection approaches, we find events not all detected by all methods. We discriminate between two types of detected events on the basis of their spatial coherency and their frequency content: Single Tremor Pulse (STP) and DeGassing (DG) events. We have introduced 2 additional figures in order to describe details of those events and show that the signatures in conventional records cannot distinguish them (Fig. 6 and 7 and Supplementary Fig. 14). Although their origin is unknown, we tentatively suggest some interpretations from their frequency content (line 333-338), their signatures on the fault zone (lines 338-340) and statistics of interevent times (line 346-359) in relation with interpretations given in literature (line 350-353).

In its current version, the manuscript emphasizes that the described DAS approach is valuable in the sense of allowing remote monitoring, which does not put researchers at risk during the cable installation. I do not follow this argument, because (1) the cable was placed closely to the crater and (2) seismometers and infrasound sensors yielded the complementary measurements to the DAS approach and were collocated with the cable. So it is not clear why DAS has an advantage over conventional sensors in terms of safety.

The deployment of any instrument in the vicinity of the craters can indeed be dangerous and this holds for any instrument. However, volcanic activity is time-dependent. An area that is safe one day can be dangerous at another time. By anticipating where resurgent activity might happen, and deploying the cable during a quiet time, the risk can be minimized, indeed this is true for any method. However, for DAS, safety is improved for several reasons (few are suggested in the discussion, lines 443-446).

1. Once the cable is deployed, it requires virtually no maintenance (except in the case of a cable break, of course), especially important in case of eruption. As the cable length can be up to 20 km or more, the interrogator can be located at a safe distance. In contrast, conventional geodetic or seismic sensors require power locally. Therefore, batteries need to be swapped regularly, and for those close to the active craters, solar panels need to be cleaned regularly from ash.
2. The large number of channels makes it possible to enhance the signal to noise ratio, therefore partially relaxing the need to get really close to the action.
3. It was not clear enough indeed in the first manuscript that records used in the published statistical analysis at Etna (Cauchie et al., 2015, reference 33) were obtained by analyzing data from sensors deployed close to the crater rim, whereas our results could be obtained for smaller

amplitude events using the cable deployed at >2 km from the rim. It was definitively safer to deploy at 2 km away from the crater rim than at the crater rim itself.

For the sake of agreeing with the reviewer's next point, we removed the reference from the manuscript. As the safety's argument is certainly not the major argument for measuring DAS on a volcano, we have made it less prominent in the text and we now just mention that we have deployed the cable at safe distance from the crater (new lines 119 and in the discussion lines 444-446). We do insist in several new places that a proper understanding of volcanic activity resides anyway in multiparametric observations and DAS is now able to bring additional light on structure and processes (new lines 49-51; 399-403; 458-460).

That said, the study is interesting and sound such that the safety argument is not needed anyway. It will be clearly of interest to a wider audience as it demonstrates the potential of DAS data for volcano monitoring.

We completely agree, thanks to highlight this statement.

The manuscript is clearly written, only some English editing is necessary.

We have checked the manuscript carefully.

Besides the points above I have few points of criticism and mostly minor remarks. Although addressing my review may require some text modifications and/or additional explanations, I find this submission is close to publication quality.

Fabian Walter.

GENERAL REMARKS

As mentioned above, the authors should better embed their findings within the existing volcano literature, especially that pertaining to Etna volcano.

We introduce a complete description of the goal of volcano seismology (lines 54-64) and events recorded (lines 63-100). We agree with the reviewer that this makes our results more integrated in volcano-seismology research.

What is the role of the explosion events and the two event types described towards the end of the paper?

Gas is the driving force for eruptions: without gas, not eruption. We attempt an over-simplified explanation: when gas can escape gently from the magma (depending on its viscosity and permeability), lava flows and gas is released gently, or with mild explosions (like in Reykjanes eruption at present). Explosions occur when the gas flux is larger than what the magma/rock permeability can sustain, and the pore pressure exceed the magma/rock strength (large explosion can then occur, such those occurring at andesitic volcanoes such as Merapi). The two events that we detected are not really described in earlier studies or even reported before. As we have denser spatial data coverage, which is a

clear benefit of DAS compared to other equipment, we could develop new or apply recent detection schemes, and the result is that we found and could discriminate those events on the basis of their frequency content and their spatial coherency. The interpretation, as noted by another reviewer has still yet to be detailed, however, we suggest and speculate that they are linked to gas movement in the volcano.

What are the source mechanisms of these signals?

The events show complex waveforms, presumably due to a combination of source and propagation effects, such that deriving source mechanisms for these events is highly non-trivial and definitely beyond the scope of our study, especially with the limited spatial extension of our cable. This will be possible when a longer cable would be deployed surrounding the craters (as suggested at line 441-443). However, we suggest they are due to fluid migration at different depth in the conduit.

What is known about the layer of scoriae?

The scoria layer is known by field terrain volcanologists, but there is no published report on it, except the geologic map (reference 28). This study is to our knowledge the first detailed report of its characteristics, thickness, velocities, and non-linear response.

Is this a failure-prone layer that plays a role in lahar or other types of dangerous mass movements?

This specific layer is not unstable, as it is on a flat area. There is no known lahar risk at Etna. We included a discussion about the fact that the infrasound associated to any (large enough) explosion may trigger nonlinear behavior at other location in the volcano, for example unstable scoriae layers of newly formed steep cones, where unconsolidated (and unwelded) volcanic product accumulates) suggesting that they could indeed be destabilized and pose a hazard (lines 429).

What is the benefit of knowing the layer thickness distribution?

The knowledge of the superficial layer allows for

1. evaluating the risk of instabilities. It is necessary, if infrastructure is planned/needed. The understanding of the behavior of such material under various solicitations, can be of interest for other regions with scoria like superficial layers (very loose and unconsolidated material).
2. brings better understanding of the recorded signals, as we can constrain models better.

How are tremor pulse events and degassing events generated and why are they relevant?

As mentioned above, gas is the motor of eruptions. As it appeared from the analysis of the DAS data, STP and DG events could be originated by the same process, i.e., fluid movements within conduits, at depth or closer to the surface. Understanding those processes is relevant to understand better how fast, how well gas can escape and where gas is located within the conduit, may be allowing to anticipate explosions and volcano unrest? Those elements are partially integrated in lines 431-436.

The sentence on Lines 264-265 seems rather weak and general and should be solidified with discussion on volcano seismology.

We agree that this sentence is not sufficient. A full presentation on volcano seismology has been incorporated in the introduction and the link of those new events discussed at lines 338-363 and in the discussion (lines 431-436).

Similarly, what is the physical meaning or interpretation of the specific detection statistics in Figure 4c?

Statistics do not preclude or exclude any mechanisms. We introduced interpretations of LP events in literature (introduction), and added our own contribution to this discussion, in light of our results on the detected event types. The overall idea is that in the absence of more direct information, the distribution of inter event times can be a clue to the causative mechanisms by comparison with literature cases. We demonstrate that our event detections techniques distinguish two sorts of events, both of them following rather long-period event statistics (interevent times), but not tectonic like events statistics. The results make them more prone to be LP-like. If indeed they are related to degassing, as suggested by the Supplementary Movie 2, then a unifying generic source mechanism could be searched and only their distance to the free surface could explain the different frequency content. Generated at depth, seismic waves would propagate in more compacted media, and then be less prone to scattering; generated at the surface the waves would be more dispersed and scattered.

For the broad audience of Nature Communications, additional explanations of these events are necessary.

We agree, and have modified the text accordingly, as suggested above.

Have these events been studied on Etna and elsewhere previously?

These events were not identified before at Etna. We introduced this terminology here. Note that this was possible with the DAS only, thanks to the density of the observation. Those events are identified thanks to the dense observations' capability of the DAS, they existed but not identified at such, hidden in the tremor. From the frequency content and the statistics, they are more LP-like, with smaller amplitude. We suggest they are related to intermittent build-up and release of strain (lines 353-355) and interpreted as a series of LP events linked to fluid pulses migrating within the conduit (lines 356-360).

Also, why were the explosions, which apparently leave a clear seismic signature on the DAS record not considered as a third event type for automatic detection?

The detection algorithm was applied to continuous data irrespectively to the event types. The event types came out from the analysis of the detected events by the different methods. We identified the different types from their occurrence within each method. The largest explosion at 10:54 was detected by the 3 methods. However, the other smaller explosions (e.g., at 14:04, Fig. 3) are not detected by the similarity method, potentially classifying them as DG event type. Explosions are detected as a DG event type, as they contain higher frequencies and are not coherent enough to trigger the similarity algorithm. There are 5 craters at the summit, which may trigger several signatures of signals, which could be detected differently.

SPECIFIC REMARKS

Abstract: infrasound is not mentioned although it plays a central role in the study.

Thank you to point this out. We have indeed detailed which are the techniques involved by replacing “conventional methods” by “seismometers and infrasound sensors”. We cannot say more in the abstract

Line 45 and elsewhere: What is meant by “disruptive progress”?

We removed those words following the editorial guidelines.

Line 51-52: How do infrasound waves reveal source depths?

Infrasound are generated at the explosion site. Reference 11 (Johnson and Ripepe, 2011) indicate the relationships between (1) the energy partitioning of the explosion into the ground versus air, (2) the depth of explosion, and (3) the amount of overburden, can be determined through infrasound signals in combination with seismic signals.

We added “in combination with seismic signals” in the manuscript to make it clearer that only IF is not sufficient (lines 87-94).

Lines 109-112: I do not understand where the maximum propagation distance of 300 m comes from.

We used the formulation indicated in the (new) reference “10” (Medicci et al, 2013), equation (6).

Citing the important sentences to understand the method:

“A spherical pressure wave propagating in the atmosphere normally decays with the inverse of the traveled distance due to geometrical spreading.... Using that inverse proportionality of the measured pressure from a sensing station

within a few kilometers of the vent, the reference peak pressure at the vent can be calculated as

$$P_k = P_m r_m / r_k \quad (5),$$

where P_m is the measured pressure, r_m is the distance from the vent of the volcano where P_m is measured, P_k is the peak pressure at the reduced distance, and r_k is the reduced or reference distance at which P_k is calculated.... most volcanic eruptions are in the range of weak shock waves.... [however], the strong shock wave model can still be used in weak shock waves, but it will not be a suitable model for waves that have decayed to sonic speeds. For that reason, it is convenient to identify when the transition between supersonic and sonic will occur. Two new quantities, the transition distance, r_t , and the transition pressure, P_t , are defined as the distance and pressure at the transition between sonic and supersonic flow.... the transition will occur just above Mach number of 1. We use a transition Mach number, M_t , in the range between 1.01 and 1.015. The pressure in front of the shock wave is the atmospheric pressure, so the transition pressure can be calculated as $P_t = k_t P_{atm}$, where k_t is the pressure ratio obtained using equation (2) for that given transition Mach number. In this case a k_t between 1.023 and 1.035 was obtained. The transition distance r_t can then be solved from equation (5) in terms of P_m as:

$$r_t = P_m r_m / [(k_t - 1) P_{atm}] \quad (6)$$

The energy that generated the shock wave during the eruption, E_s , can be calculated taking advantage of the pressure versus distance approximation for strong shock waves developed by Taylor [1950]. The energy released by the eruption will power a shock wave that will decay into sonic wave at approximately the distance, r_t . Since this is the maximum distance that a shock front can travel for a

given initial detonation energy, the energy released can be obtained in terms of r_t and P_t or in terms of r_m and P_m using equation (6):

$$E_s = P_t r_t^3 / 0.155 = (P_m r_m)^3 k_t / [0.155 (k_t - 1)^3 P_{atm}^2] \quad (7)$$

We recomputed the values, and indeed there was an error (typo?) in the values computed: it is not more than ~100 m (at 2800 m elevation). Thanks for the accurate checking. We also incorporated the method qualitatively in the introduction, to make our estimation clearer.

Paragraph starting on Line 114: If I understand correctly the authors refer to wave polarization relative to the cable orientation when arguing that segment B2 should record higher amplitudes than segment B1. On the other hand, B1 is closer to the source. Is this relevant?

The understanding is correct. However, B1 is not really closer to the sources, but the corner between B1 and B2 is. For both segments, the portions of the fibre closer to the source (in both B1 and B2), amplitudes are indeed slightly higher, but the wavefield polarization has a stronger effect.

Discussion of Figure 2: It may be worth pointing out that the infrasound frequencies shown in Figure 2a do not show up in the cable (Figure 2h). So the infrasound signal in the DAS record is really an induced signature rather than a result of cable-air coupling.

Thank you for this assessment. We introduced a sentence referring to this (lines 228-230).

Line 148: I would refrain from the word “perfect” for any type of measurement-based evidence.

We agree. “perfect” replaced with “excellent”.

Lines 156-157: The higher frequencies are suppressed rather than absent.

We disagree. The high frequencies cannot be suppressed as they initially do not exist in the exciting signal (infrasound waves). This is the whole point of the “piano concert” where we demonstrate that if high frequencies would be present in the atmosphere they would be recorded by the infrasound sensors. They are really absent, except where the nonlinear interaction of the infrasound with the ground generates them.

Line 16: “material of 1 to 3 cm diameter” \diamond “grains of 1 to 3 cm diameter”

We agree, changed (at line 160, not 16 😊).

Paragraph starting on 165: The shift in resonance frequency peak is the most prominent feature of Figure 2. The derived thickness and variations thereof should be compared to previous studies or alternative measurements (e.g. H/V ratios or autocorrelations of seismic point measurements).

We are not aware of any other studies at this location with H/V ratios or any other techniques. We have therefore performed H/V computations from 1 hour of geophone and broadband stations data. We do not identify any frequency peak at 15-25 Hz, rather the main peak lies at about 0.5 to 1 Hz and is not the same for all geophone positions. In theory, H/V peaks may have 2 origins as reported by Bonnefoy-Claudet et al. (2006). 1. horizontal polarization of Rayleigh waves is well developed (superficial sources are far enough), and Rayleigh waves can develop; 2. the amplification ratio between S- and P- waves (for

volume waves and if sources are deep). In our case, the seismic waves are generated by infrasound, which would correspond to very local sources and Rayleigh waves cannot develop. This may explain why conventional H/V ratios on ambient noise cannot see the observed frequencies (Fig. 2). We trust that this beyond of the scope of our paper, and therefore have not included this in the manuscript.

Bonnefoy-Claudet, S. Cornou, C., Bard, P.Y., Cotton, F., Moczo, P., Kristek, J. & Fäh, D. H/V ratio: a tool for site effects evaluation. Results from 1-D noise simulations. *Geophysical Journal International* **167**, 827–837, doi: 10.1111/j.1365-246X.2006.03154.x (2006).

Paragraph starting on Line 202: For the location and the azimuth determination of the fault some more evidence should be given, ideally with a supplementary figure.

Done. Figure Supplementary 12.

Paragraph starting on Line 293: This contains absolute statements such as “first time” and “unprecedented accuracy”. To my mind this deserves justification in terms of 1-2 discussion sentences.

We have removed those terms, as they are indeed not in the vocabulary recommended by Nature Communications.

Line 312: “Weak” is a weak qualification. With respect to what?

We agree. We changed to “We showed that signals with low signal/noise ratio can be ...”

Line 314: which conduit?

We added “volcanic”

Line 315: “information density” is unclear.

We replaced by “density of information”

Lines 318-320: The DAS cable has to be installed, too.

Indeed. We have explained it better.

Lines 322-325: I find these statements rather generic for a conclusion. Underground deployment of a cable seems laborious compared to geophone installations. The big data paradigm is generally applicable to DAS and many modern seismology applications. I would stay a bit closer to the findings of the present study and open up the perspective from there.

We have reshaped the discussion and extended the argumentation with that respect as well (lines 463-469).

Line 533: Throughout the text I was wondering if the cable was there first to transmit seismometer data or if seismometers were used to magnify the cable installation. This would be a good location in the text to clarify this.

There is no hierarchy in this design. We set-up the cable with both aims, benefiting from the two features a multi-fibre cable offers.

Line 557-559: Can the loop effect be highlighted better?

We added “loop” in figure 2, where the time signal and frequency are similar within the traces included in the loop.

Line 593: Where does the bandwidth for the propagation velocity come from?

The apparent velocity is higher than the true velocity, because it has an incident angle with the cable. If the wave is propagating parallel to the cable, the apparent velocity is the true wave velocity, and is it the lowest value. If there is an angle, the apparent velocity is higher than the true velocity. The apparent velocity has been computed at geophones and BB sensors location using the method “DAS strain validation: Phase velocity”. We compared the amplitude of the DAS strain with the seismic particle velocity. We find that the apparent propagation velocity c varies along the profile, as reported in Supplementary Fig. 4a for the BB sensors and in Supplementary Fig. 4b for each geophone.

Line 601: Which “two displacement vectors”?

Let's assume A and B are two locations of the cable distant by dx , e.g., two geophones locations. Under strain due a seismic wave, let assume A moves to A' (which defines a displacement vector $\overrightarrow{AA'}$) during a time interval. During the same time interval, B will move to B' (which defines a displacement vector $\overrightarrow{BB'}$). $\overrightarrow{AA'}$ and $\overrightarrow{BB'}$ have no reason to be the equal, e.g., because the seismic wave is not in phase at the two locations. The difference between the two displacement vectors $\overrightarrow{A'B'}$ is defined as the difference du between the two displacements. This basic formulation is well described in the reference 55 (Segall, 2010).

Line 603: Rewrite “avoiding the knowledge”.

We agree. Rewrote.

Line 610: “better agreement” should ideally be quantified.

We quantify the agreement with the normalized RMS error for each time series. We added the averaged RMS error and the median for each method in Supplementary Fig. 4.

Also, the following paragraph requires a reference upon which the numbers and mathematical expression are based.

We do not understand which “following paragraph” is meant here to be missing a reference. The mathematical expression in paragraph line 539-540 refer to the reference 57 (Bodin et al., 1997). The following paragraph (with equation (1)) refer to the reference 58 (Wang et al., 2018).

Method 3: It would help the reader to be reminded that you are referring to seismic waves transmitted into the ground by infrasound signals.

We indeed are not referring to neither the infrasonic wave or the infrasound induced seismic wave, but the propagation of the seismic wave component of the explosion, neglecting the infrasound signal, which is not modelled here. We added the word “seismic” to make this point clearer.

Line 656: Why is the explosion source modeled as a single force (rather than an isotropic moment tensor)?

This is a good question. There are several ways to model explosions, discussed in the literature. As it is at the free surface, it can be assumed to be a single vertical force. For completeness, we performed an additional computation with an explosive isotropic source. The results are very similar (except amplitudes). We have added a line with this computation performed in the complex case. This shows once again that we cannot perform inversion with only this array.

Lines 704-795: rewrite “records being toward ...”

We changed the expression.

Line 709: Why “successively”?

We removed “successively”.

Line 737: Equation (3): why are there such large spaces in the equation?

This is a typo in the equation, corrected.

Line 758: overpassed \diamond exceeded

We agree, changed.

Lines 782-783: “most meaningful for set of channels ...” not clear what this means.

Rephrased.

Lines 788ff: The permanent volcanic tremor is an entirely new concept. Is this another source type? Also, the sentence starting with “The permanent ...” should be rewritten.

We now describe more thoroughly the different seismic signals recorded at Etna in the introduction. Therefore, the concept described here is actually not “new”. We rephrased the sentence.

Lines 987-988: I would avoid accumulation of more than three nouns.

We removed one noun.

Line 1022: “spectrogram with time”: unclear

Indeed! Corrected.

Line 1059: stroke \diamond struck

Corrected.

Lines 1065-1066: “systematic differences in the amplitude with depth”: unclear

Corrected.

Lines 1078ff: how were shear forces and yield strength calculated/determined?

Shear forces were calculated based on a simple elastic model for the optical cable as described in Reinsch et al. (2017). Based on the design of fiber optic cable as well as the elastic properties of the multilayer cable, the force needed to elongate the optical fiber, hence to generate a detectable signal, can be calculated. The overall stiffness of the cable is determined by the stiffness of its individual constituents. As external forces are acting on the outer layer of the cable, shear forces at individual interfaces (with specific surface areas per unit length) transmit the forces towards the optical fiber. As the theory is presented in reference 89 (Reinsch et al., 2017), we did not include more details here. We acknowledge, however, this valuable comment and added a short explanation. In order to address the comment on the determination of the yield point of the material, we refer to industry standards for producing fiber optic cables and also refer to (Reinsch et al., 2017) and references therein, where typical gels have yield points of 34-140 Pa. For the gel used in this specific cable, the manufacturer provided viscosity values for shear rates of 50 and 200 s^{-1} . Assuming a Bingham fluid, we calculate a yield point of 60 Pa that is much higher than the shear forces applied at the interface. Hence, we assume elastic behavior of the gel. We added a line in the text accordingly.

Lines 1088ff: What does reorganization mean? Sliding of grain boundaries? Also, “are likely to provide sufficient” sounds clumsy.

Thank you for this. Reorganization indeed means sliding of grain boundaries. We edited accordingly. “are likely to provide sufficient” is now been replaced.

Line 1107: “was not be recorded”: rewrite

Done, thanks!

Lines 1120-1121: permanent frequency \diamond sustained resonance

Changed.

Line 1128: “following the deep cable backward” is unclear

We fully described the cable layout into figure caption 1 (lines 148-158), which also fulfills requirement from other reviewer (“not clear how the cable is laid out”).

Lines 1132ff: “seems rather very long period” sounds clumsy. Same for “matches approximately well”.

Both rephrased.

In several parts of the text the authors use the term “elongate”. Is this used synonymously for “stretch” or “strain”? I would keep the terminology as limited as possible to avoid confusion.

“Elongate” is used 3 times. “Stretch” is used 1 time. We changed all “elongate” occurrence to “stretch” as the latter term is more general, and as the fibre can also be shortened.

The piano tests receive a lot of attention. Why is the video necessary? Tuning a piano should be a familiar concept to all readers requiring little explanation. The supplemental figure suffices to my mind.

We wanted to demonstrate that infrasound sensors were close to the piano. We removed the video.

Video “Explosion at NSEC on 5th September 2018 at 10:54” cannot be opened.

This is unfortunate. Sorry for that. The format is avi and is working well under both Windows and Linux exploiting systems with VLC software.

FIGURES

Figure 1: A scale bar for the bottom right inset is needed. What is the yellow box in this inset?

We have added one scale bar in the figure, and an explanation of what the yellow box is.

Figure 3: It would help to see another box in the (a) panels showing the zoom-ins of panel (b).

Fig. 3 is now Fig 4. This is actually indicated in the 3rd panel of figure 3a, as the little blue box. We included an explanation in the caption – which was indeed missing.

Figure 4: The y-axis label should be defined in the caption.

Fig. 4 is now Fig 5. Indeed. Done.

Extended Data Figure 3: Define y-label in upper panel.

Ext. Data Fig 3. Is now Supplementary Fig. 10. Indeed. Done.

Extended Data Figure 4: Upper panels: The “(2 m)” in the x-labels is confusing. It sounds like 2-m-high jumps. “MASW” should be written out once in the caption (same for acronyms in other figure and table captions).

Ext. Data Fig.4 is now Supplementary Fig. 11. From the picture in Supplementary Fig. 3, one can see that the jump was indeed not 2 m high 😊. We removed “2 m” and explained in the caption what it refers to. We have also explained MASW in the figure caption and checked all figure captions.

Extended Data Figure 6: I could not verify the last two sentences in the caption and am not sure I understood the content and how to compare model and data.

Ext. Data Fig. 6 is now Supplementary Fig. 7. We have included more explanations to make this point clearer. The order of magnitudes is indeed different between the different models, as seen in the color scales. The order of magnitudes obtained with the fault zone, the topography and the shallow low velocity layer produces amplitudes that are similar to the observed ones (Fig. 2).

Supplemental Figure 1: I would label the notes in the upper panel.

Supplemental Fig. 1 is now Supplementary Fig. 8. Good point. Done. We also added the theoretical values of each note played.

Supplemental Figure 3: I cannot verify the resonance at 25-30 Hz. Rather, a concentration of spectral energy?

Supplemental Fig. 3 is now Supplementary Fig. 13. Corrected.

Reviewer #2 (Remarks to the Author):

Review to the manuscript: "Fibre optic distributed acoustic sensing of volcanic events" by Jousset et al.

The authors deployed a Distributed Acoustic Sensing (DAS) on Mount Etna, Italy and demonstrate the potential of this "new" technique to detect and analyse volcanic events as well as imaging volcanic structural features. The authors compare the results with a set of broadband, geophone and infrasound sensors. They provide videos, which made the manuscript pleasant to discover. The DAS is applied for the first time on a volcano, for the positive data resulting from the survey and the promising future applications of this technology, I recommend the manuscript for publication after taking into considerations the following comments.

Thank you for those encouraging comments.

The comments:

Lines 121-123: Using 3D wave propagation simulations, we demonstrate that Etna topography and local volcanic structures explain the wavefield variability of both particle motions (Extended Data Fig. 5) and strain rate amplitudes (Extended Data Fig. 6).

It seems from these two reported figures that you show the simulated results from modelling, but do not present a proper comparison, where I would expect differences between observed and simulated data as well as the computation of a misfit measure.

The objective here is not to perform full waveform inversion, which would provide the best way to make sure the model suits best data, in terms of arrival times, amplitude of the signals, etc., indeed required for quantifying the explosion amplitude. This is a very valid objective, but we did not attempt as our array is clearly not suited for this. Therefore, presenting the waveform misfit will hardly make sense. However, we provide amplitudes obtained with the different simulations. Amplitudes are most appropriate in the modeling accounting for the fault zone, topography and low velocity shallow layer.

For Ext. Data figure 5, enhance in the caption or in the graph itself that points a-e are for synthetic results and f is for observed data. For Ext. Data figure 6, observed data should be added aside or below to better see how the data can be compared.

Ext. Data Fig 5 is now Supplementary Fig. 6. In order not to confuse the reader and not overinterpret our synthetic models, we removed all direct comparison with data. Instead we introduced a further simulation as suggested by reviewer 1.

Line 129- Fig2: This is not easy to understand on Fig 2g and h where the channels are located, in reference to Fig 1.

In order to make this clearer, we have indicated geophones labels at their position along the cable.

Why is B2 inserted into the profile B1?

B1 and B2 are comprising both deep and shallow cables. The cable has first a deep section along B1 branch with channels 1 to 410, then the cable turns (still deep) along B2 with channels 411 until 520, then the cable has a shallow section (above the deep cable), with channels 521 until 630 (with same geographic location as deep channels 520 until 411, respectively), and finally, the shallow cable turns

along B2 (still above the deep cable), with channels 631 until 715 (with same geographic location as deep channels 410 until 315). We have introduced this explanation in the caption Fig. 2.

Lines 228-243 Fig. 3: As you also mentioned in the caption, the theoretical strain rate arrival times are barely visible on the subplot c. This could be brought out with different colors and with an onset on a particular time interval.

Fig. 3 is now Fig.4. Thank you for this good suggestion. We have plotted the theoretical arrival times for each crater in figure 3c in an inset where we see with better resolution the different theoretical arrival times. However, as BN, VOR and NEC are in the same azimuth with respect to the cable, we cannot distinguish them.

Line 241: Is the explosion seen in supplementary video 2 associated with the records in this Figure 3? If so, the time occurrence is not consistent 10:54 for the signals and 11:00:42 for the small degassing. Would it be possible to add a time scale on the video and/or the signals as in supplementary video 1?

This is a typo. Sorry for this. The video we referred to line 241 (old manuscript) is NOT video 2, but video 1. Note that in Video 2, we cannot associate the time as rigorously as in video 1, as the video 2 was taken on a phone by a guide on the crater rim (by chance!).

Line 252: Extended Data Fig 7: could you add the signature of the strain rate for the two types of signals in this figure or in the manuscript? (As in Fig 2a)

Ext. Fig. 7 is now Fig. 6. Thank you for this suggestion. We have extracted a portion of the Fig. 6a and plotted it in Fig 7. This helps indeed to identify clearly differences between the two events. In addition, we included raw data in Supplementary Fig. 14.

Lines 260-270: statistical analysis: Was it performed without distinguishing the two types of events (STP and DGE)?

Yes. We did not classify the events before performing the detections. We may wonder whether there is a mixture of STP and DG event occurrences in the counts for each method, which then would indeed bias the statistical analysis and make the results artificially similar. However, we observed, as shown in Fig. 6 and Fig. 7 and Supplementary Fig. 14, that the events are indeed different. The STP are best detected by the similarity method, and the DG event are not detected by the similarity method. Reversely, the STP are hardly detected by the STA/LTA method. Therefore, we are confident that our results are representative of the statistics of their own occurrence, and that mixture between the two is marginal.

Figure 4b and c show similar information, Figure b could be moved to supplementary material.

Figure 4 is now Fig. 5. We did not move figure b to the supplementary information. Instead, we have better explained how we obtained our results (lines 346-353) and added a method: "Probability Density Functions of interevent times". This provides the basis for discussing possible mechanisms behind (lines 354-359).

What is the importance in the manuscript to show the results from the 3 types of detection method?

The events we are dealing with here are very weak and hidden in the tremor ($5e-9$ strain rate, see Supplementary Figure 14). As indicated in the first point of this comment, it is important to have a separation between events to classify them. It appears that if we would have used only detection method (STA/LTA), as initially performed during our study (not reported here), we did not detect all events, some were obviously left over. Therefore, we tested additional methods to try to detect as many as possible. After tuning the frequency range, and several tests, we finally obtained shown results. Note that it is possible that the observation that very small inter-event rates are rare is an (unavoidable) artifact of any detection algorithm, explaining larger misfits at low interevent times.

Was also the amplitude of the STP and DGE investigated?

Indeed, we do not report statistics of relative amplitudes between events.

What about the location of those signals? Are they all coming from the same source? Does it show activity in the different craters or localized to one in particular for the investigated time interval?

These are good questions. However, estimated locations of those signals with one array only would be highly inaccurate. However, two possible way to start obtaining hints could be:

1. search for back-azimuth for those events, however, the local structure distorts the seismic waves and with only one array it would be very much inaccurate.
2. compare waveforms between events and classify in families. We did not attempt this, but clearly this would be a future step.

291 Advantages/ disadvantages of the DAS should be discussed more in details in this conclusive paragraph : limitations/installation set-up/ what could still be improved for further studies?

Thanks for this comment and suggestions. We rewrote the discussion including limitations and installation set-up issues (line 438-451).

Supplementary material: The structure of the supplementary material (methods, notes and extended data) should be improved. This would be easier for the reader to have, in the manuscript, the reference to one supplementary note/file/figure and not several as for the example: line 1585 (supplementary Fig. 1, Supplementary note 2 and supplementary video 3). At least supplementary information should be gathered. One could in the manuscript refer to the supplementary notes, that in turn refer to the associated figures.

We have reorganized the whole structure of the manuscript to fit with requirements of Nature Communication.

Also, attention should be taken for introducing the note/figure/table references in the order of appearance:

ex line 106: supplementary Table 3 is mentioned before supplementary table 2

ex lines 123-124: extended data Fig. 5 and 6 before extended data figures 2-3-4.

As it is, it makes the reading a bit confusing.

Thanks for this. We took care of the right order of materials.

Reviewer #3 (Remarks to the Author):

Review of manuscript “Fibre optic distributed acoustic sensing of volcanic events” by Jousset et al. for publication in Nature Communications

The manuscript shows an initial application of Distributed Acoustic Sensing (DAS) to volcanic environments using data from a cable deployed within a few km distance from the summit crater of Mt. Etna. Results are presented for (i) subsurface imaging using resonances caused by volcanic explosions (this though turns out to be simply estimating the scoria layer thickness, which is not very exciting), (ii) wavefield recordings of seismic and infrasound signals and their separation, and (iii) detection of volcanic signals (monitoring). The goal is to show “DAS as a new tool for volcanology”, the title of the discussion section.

DAS is a new technology that is starting to be used in a wide array of applications. The optical cables measure strain rate and the cable acts as a dense linear network of strain sensors. That is really cool and offers opportunities for wavefield monitoring and the detection of small signals (through stacking or other signal enhancement strategies) previously not (or basically not) available. DAS will play a significant role in geophysics (seismics, seismology) and volcanology. The lead author (and some of his co-authors) have published a similar, initial study introducing DAS as a tool for imaging of seismological and structural features (Jousset et al. 2018, Nature Comm. 9, article 2509) and this manuscript is trying to do the same for volcanology.

The example applications show the usefulness of DAS. No question.

Thank you for those factual comments.

What they do not show is was DAS necessary to obtain them or did DAS make it easier to reach the results or are they better constrained? It seems, the same data analyses could have been performed with seismometers and/or infrasound sensors (networks, small arrays) at the same ‘safe’ distance from the craters.

As mentioned later by the same reviewer, it is very challenging to demonstrate at the same time the benefit of a new technique, if it not validated with conventional methods and known observations. Would we have shown DAS results without seismometers and infrasonic signal validating them, we would presumably have been challenged on their validity. The whole method section about the validation of the strain is there to demonstrate that indeed DAS records provide similar signals as conventional sensors, which is good news! However, the real benefits of DAS were possibly not clearly described. We have introduced several points in the discussion/conclusion highlighting some key features:

- Better understanding of structural features, by comparing differences between successive traces thanks to the much denser spatial sampling
- Strain rate and strain are more sensitive to local features
- Wavefield propagation is better understood
- Signal distortion near faults
- Ability to perform wavefield separation.

We could do this by deploying a large number of seismometers located every 2 meters over 1.3 km, which means 750 seismometers, batteries, GPS antenna, datalogger, etc... the amount of work for deploying, maintaining and data downloading and management with such number of sensors would be

very intense, and we claim this was much simpler with a single cable, which is by the way still in place and can be reused at any time, by just bringing an interrogator, plug and play. Those arguments are included in the discussion (e.g., lines 438-451). In addition, we have detailed results on the small events and their difference by introducing additional figures (Fig. 6 and 7 and supplementary Figure 14). We show that features found in the DAS data, cannot be obtained in sparser conventional observations (Line 359-362).

The manuscript repeatedly mentions that DAS may make studying a volcano a safer undertaking but I do not see why (lack of a compelling argument). I did find the use of an obituary, to say the least, a little disturbing when talking about safety and I would hope the authors could come up with a better argument. DAS will not replace or aid geochemistry (and gas measurements).

The safety argument has been removed and replied at length to editor and reviewer 1. We mention it still only in the discussion and we removed the reference. In addition, we claim that DAS can complement any other observations (lines 432-436).

The manuscript is well written. The sections Extended Data and Supplementary Information contain a few more grammar and word choice issues than the rest of the manuscript and should be looked at again; sorry I'm not providing a list (L758 'overpassed' do you mean s.th. like 'exceeded'?)

Corrected.

L941 'profile C672bwd', should this be 'profile C672bkw'?

Indeed, corrected! Thanks.

There are a few more small things like that)

We tried to correct most of the typos.

The authors used an impressive range of analyses methods and I have no significant issues with the methods.

(i) I found the wavefield separation example particularly interesting. I'm fairly sure though, that infrasound data by themselves suffice to determine the source crater for the explosions?

Infrasound are indeed the best method to locate surface explosions. The interesting point is that the infrasound sensors could not find the nonlinearity of the subsurface and yet the propagating response of the wave allows also the fibre to be used to locate the source as well. In addition, infrasound alone cannot detect the small events as demonstrated in Fig. 7 and Supplementary Fig. 14 and reported in the discussion.

(ii) The argument about the need to use Etna topography and possibly a fault zone and low near surface velocity is not particularly convincing (Extended Data Fig. 5)

The features used in the modelling are facts from the field. The topography of volcanoes has been shown to introduce distortion in seismic wavefield – Etna topography is rather prominent – the faults zone are features known from the geology and also visible from the DAS observation (there is an

increase of amplitude in the strain rate data as shown e.g., in Fig. 2 and 3). The model is as any modelling an oversimplification of the reality. However, both travel times and amplitudes fit better with topography, fault zone and shallow low velocity layer included. In addition, the complex wavefield is better reproduced at least qualitatively with complex structure.

and I wonder whether showing particle motions (of what exactly? For Flat Topography is that simply P and S or also surface waves? Would be good to explain what's shown for the simplest case); most obviously the horizontal component motions do not work (flat earth synthetics vs. observed) but none of the more complex models really does a good job either.

We improved the presentation, by adding Supplementary Fig. 5 with synthetic waveforms of the modeled signals. Amplitudes obtained from the similar source (source amplitude obtained from the infrasound data) for the different models are not the same (10^{-7} s^{-1} for models a,b,c; 10^{-5} s^{-1} for models d and e). Those amplitudes fit better with observations when structural features are included. We describe better the results (line 183-189). Neglecting the existence of the fault and the shallow low velocity layer does not make the modeled results closer to the observations. We insist that those results are indicative only of the influence of the subsurface role for strain measurement using an improved forward modelling approach (up to now very few dynamic strain modellings exist). Performing an inversion with only one array, would clearly be much more unconstrained, and results would be unreliable.

(iii) The DAS monitoring part detected two types of recurring signals that were detected with STA-LTA and stacking, and with 'local similarity', respectively; it is not clear what these signals are and where they come from; the authors seem to postulate one is a (deeper) tremor pulse event (lower frequency) and the other a shallow degassing event (with higher frequency signal) attributed to 'a ... expression of fluid movement'. A similar temporal behavior of these signals and of LP events makes the authors suggest the two event types and LP events might be 'somehow related.'

First, was DAS necessary to find these two types of signal?

The detected events are visible as tiny vibration in the seismic records, but they indeed start to make sense from the DAS records, i.e., they can be classified as STP or DG events from the ability of the different detection methods to detect them within the tremor (Figs 6 and 7 and Supplementary Fig. 14). Spatial coherency characterizes the detection. DG events are detected with STA/LTA whereas the STP are coherent spatially and can be detected on the similarity algorithm only. It is pretty clear that with single stations we cannot extract features seen by the DAS, even located at the same location of the DAS cable. In order to detect and understand such events, closer and numerous stations in a large-N deployment would have to be used, with much more time and effort. We added such argumentation in the discussion as well (e.g., lines 359-362).

Second, I don't follow why similar temporal behavior statistics would lead to a relation between these signals and LP events.

We have introduced a better reasoning of the argumentation (lines 346-363) and introduced a new method: "probability density functions of inter event times".

Third, 'whatever the interpretation ...' simply shows that DAS is not a panacea, somebody still has to come up with a viable interpretation for 'whatever' signals are being recorded.

This is indeed a valid assertion. We did not claim that DAS is a panacea, in fact no technique alone is able to understand the complexity of volcanic phenomena. Only a combination of observations can bring full understanding of volcanic phenomena (e.g., Surono et al., 2012, reference 2). We only claim that DAS can bring its share (as recognized by all reviewers and the Editor), and we claim indeed that DAS leads to novel observations. We easily observed and identified tiny events from raw DAS data, which were not previously reported as far as we found out, because the observation methods used so far simply could not identify them. We propose an explanation, that need to be challenged, modelled, confirmed or discarded. We introduce in the text those considerations in the discussion part (431-436).

— I do understand that it is difficult to show ‘DAS works’, which requires to compare with other data that record the same signal opening up critique along the lines of ‘DAS is not needed.’

This is indeed a tough dilemma. Thanks for recognizing it.

I think what would help (me) would be a clear example that DAS shows ‘more’ (and smaller) signals of the same type as seen with other methods and/or that DAS recordings have a higher resolution than other methods (or something similar). The added gain from DAS currently does not come through clearly.

We have made those elements more clearly, by introducing Fig. 6 and 7 and Supplementary Fig. 14. Those figures and the lines 327-363 and the discussion account for all remarks from the reviewers, and we hope that the added gain from DAS come through more clearly.

Operating a fibre optic cable seems to require substantial infrastructure such that the authors conclude that a cable might be connected during a crisis rather than being operated as a monitoring tool. The huge amount of data (storage and processing) currently makes it not feasible to operate DAS continuously over an extended period of time.

There are already several places in the world recording continuously DAS data for monitoring purposes. For example, the Foresee project in urban environment started record in 2019, and is still recording as seen in this publication. <https://se.copernicus.org/articles/12/219/2021/>

These issues will be solved in the future, but for the present and near future fibre optic cables will be a research tool. And a research tool available only at a very few volcanoes.

Thanks for the reviewer to indicate what can be anticipated in the future for DAS in volcanology. We are also very optimistic.

The authors could have made this more clear, hinted at future ‘monitoring’ applications and given a few specific research examples to be addressed with DAS?

We have introduced comments in the discussion, in order to explain expectations for volcanology in the

future. We believe that the future for volcanology is open for DAS, in any case in combination with other techniques, as part of a multiparametric observation system. We have listed possible targets (concisely reported in the discussion of the manuscript):

- Introduce machine learning methodologies concepts to detect, classify, locate the various signals recorded on a volcano.
- Big Data analysis of seismic (& infrasound) wavefields; affordable large N+T seismology
- High-resolution imaging structure and its changes
- Real-time process monitoring with machine learning (“live” subsurface videos)
- Multi-array dark-fibre measurements in endangered villages / towns
- Combined seismic and infrasound (solid earth and atmosphere) inversions
- Testing larger distances to the crater to play on the safety argument, do an experiment on an instable scoria layer, combining with DSS and DTS and chemical sensing for detection of (hidden) permeable faults, ...

—

L24. ‘Disruptive progress ...’ - This is correct but many readers might be confused by the word ‘disruptive’ rather than ‘innovative’ or ‘groundbreaking’ or similar.

Changed, this sentence is actually removed.

L47. ‘... have been efficient to ...’ - what is meant by ‘efficient’?

Changed with “able”

L42-59. The paragraph suggests that DAS recordings might lead volcanologists ‘to accurately anticipate subtle, yet decisive fundamental volcanic processes’? That might be a tall order. DAS, more likely (?), will become at some, well-funded volcano observatories another tool to probe a volcano, and different datasets (incl. DAS) combined might get us ‘to accurately anticipate subtle, yet decisive fundamental volcanic processes’? I would appreciate if the authors would tone down their statement.

Those lines were not referring to the ability of DAS to be able to detect “subtle, yet decisive fundamental volcanic processes”. We are fully aware that no technique alone can make better than any other for volcano monitoring. Multiparametric observations of volcanic systems help us really to better probe volcanoes. As indicated in lines 110-111, we “extend our capability and sensitivity of deciphering volcanic phenomena” (with DAS), which means only indeed as indicated in the discussion “DAS offer a complementary tool” to probe a volcano. There may be a long way before we reach the possibility at detecting in situ nanolite growth in volcanic melt (reference 22) with any instrument deployed on a volcano actually. We have added in the discussion advantages and limits of DAS for volcanology.

Not sure I understand what the red bars in Supplemental Fig. 64 show. Times when the cable was ‘on’ and connected to the interrogator, I presume?

The reviewer means Supplemental Fig. 4 we believe, which is now Supplementary Fig. 1. We added text in the caption to explain better what we see.

REVIEWER COMMENTS

Reviewer #2 (Remarks to the Author):

Second review of the manuscript "Fibre optic distributed acoustic sensing of volcanic events" by Jousset et al.

The revised version of the manuscript and supplementary material have improved with its new structure, so facilitating its reading. The authors answered the comments provided after the first review.

I however recommend to take into considerations the following comments for a further improvement. Page and line numbering refers to the version of the manuscript without correction tracking and comments are listed in order of appearance.

Page 2 Line 60: remove the term 'injection' and only use 'transport' of magma, as "injection" is frequently used in anthropic induced seismicity (e.g., exploitation of geothermal reservoirs).

P3 L63: State of the volcano unrest -> state of the volcano or volcano unrest

P3 L73: LP and VLP are not always observed on volcanoes, understanding their mechanism is important to understand the processes occurring within the volcanoes and these signals might be considered as further elements in assessing the state of the volcano.

Please rephrase, as VT and tremor are instead the signals commonly used for early warning. Or specify the particular case of Mt Etna.

P3 L88: Rephrase the sentence in lines L88-L89, not clear in the context.

P4 L95: could you list the "fundamental eruption source parameters"?

P6 Fig1: please add B1 and B2 in legend. Another sketch in supplementary material could help the reader to better image the fiber configuration, as channel numbers are often used in the manuscript.

P7: The videos were not available for this review. Be careful to insert them again.

References are not inserted appropriately in this page, could you for instance rephrase in order to better understand the equations 5,6 and 7 of ref 10.

P7 L175-176: sentence not clear: rephrase with something like to 'In the DAS records, we analyse 2 sequences allowing us to determine the ground response and subsurface structure'.

P7, L183: FZ is mentioned several times in manuscript, but is not on the map.

P10, L236: Fig 10 and 11 show velocities ranging from 200 to 600 m/s. Where did you detect Vs up to 1100m/s? can it be illustrated in supp material too? Please then ensure the deposit depth is then correctly estimated.

P10 L239: in a few lines you have the two different concepts resonance frequency and frequency resonance. Please rename one of them for better understanding.

P16 from L346: part of references were shifted and have to be corrected as #32-> #33 and #33-> #34.

P29 L613-615: This comment is more general on the estimations of velocities of propagation throughout the manuscript. With the different methods, you mention that lower velocities are

observed in fault zones: could you please quantify?

In DAS strain rate and strain validation, you obtain c_x varying between 400 and 1110m/s mentioning that are 'upper limit values of the true propagation velocity'. Could you explain?

P30 L630 This sentence is not very clear. For the inversion, do you parametrize the model space with layer thicknesses and shear wave velocities? The a posteriori probability function is a function of the model parameters, so its sampling should refer to thickness and shear wave velocities (and not phase velocities that are your observations).

Reviewer #3 (Remarks to the Author):

Review of revised manuscript "Fibre optic distributed acoustic sensing of volcanic events" by Jousset et al. for publication in Nature Communications

First, I appreciate the detailed replies to my initial comments and the attempts to incorporate some of the suggestions. Thanks for highlighting some of the key features in the discussion/conclusions; providing such add'l information/explanation of the advantages of DAS is what I was hoping to see. I think this strengthens the arguments 'pro DAS'.

I have three remaining/new issues:

(1) I really do not see the point of adding a long intro about 'volcano seismology' (L56-L107). The writing is a bit repetitive (L61-63 and L72-74) and the introduction is now more difficult/convoluted to read (for me). The entire purpose of this summary — it reads like a review of phenomena — is not clear. Please edit and relate these features more directly to contributions from DAS (focus). I understand the add-ons are in response to reviewer 1 but they are so general that it's difficult to see the relevance for the manuscript. The introduction now seems weaker and less to the point. The point is you are introducing DAS to volcano monitoring and here give some examples to show its potential and usefulness.

(2) The text has been changed significantly in several places (particularly in the introduction and discussion parts). The style is now less smooth than the initial version and I would encourage the authors to go through the text with that (smoothness of reading) in mind. I do not have a specific example, sorry, but the parts that were changed (and/or are new) are usually more difficult to follow than before.

(3) I appreciate the reply about the wave field simulation. However,

(3.a) I do not see a point in the particle motion plots. They are shown for simulations but not for actual data. How are readers supposed to know/appreciate that more complicated models produce particle motions that 'better' follow the actual data when there's no example from the actual data? I do not see a reason to have Supp. Fig 6. considering current text and other visuals.

(3.b) Need for topography, fault zone, shallow low velocities as postulated. Supp. Fig. 5 b and c are basically identical (except for amplitude). That means the effect of the fault zone is negligible? Why not? Supp. Fig. 7 implies that the fault zone has a large effect on the strain rate as the results for cases b (topography) and c (topo + fault zone) look vastly different in

the fault zone (FZ) region. How is that possible if the traces (Supp. Fig 5 b and c) in the FZ region are very similar? This is not at all clear.

(3.c) L183-187 of the main text, “Using 3D wave propagation we demonstrate that ... topography and local volcanic structures explain the wave-field variability of travel times, ... particle motions and strain rate amplitudes”. I don’t see how the modeling demonstrates this. Are you expecting the reader to translate Figure 2g into time series or simply compare Figure 2b with the Supp Fig. 5 traces? Or are you suggesting that Supp. Fig 7e looks the most similar to the actual data (Fig. 2) from all the model runs? This is not clear. And which parts of Fig. 2 are close to Supp, Fig. 7? Would the traces of geophone C671 located in the fault zone (instead of CAZG as shown in Fig. 2 b) resemble the simulation results (it should somewhat, based on the text)? Why not show C671?

(3.d) Supp Fig 7. (SF7) (i) I don’t understand the caption. Cases e and f produces similarly ‘complex’ waveforms (Supp Fig. 5) and have similar overall timing. For a given source strength, amplitudes for cases e and f differ, but why not simply make the source stronger for f to make the signal larger (what is the argument against that?). (ii) What do you mean by “more than 2 seconds”? Supp Fig. 5 clearly shows arrivals <2 s for all examples (including preferred case f) and more specifically there’s no difference in timing between cases d-f? (iii) Looking at Supp Fig. 7 (and ignoring the absolute amplitudes), cases d, e, and f look very much alike. That means that the most important feature for the modeling exercise is the introduction of the shallow low velocity structure. Why not acknowledge that? (iv) I am also surprised about the strong signal in the ~250-300 channel range considering that the fault zone is near C671 that is near channel 340 or so; or did you move the fault zone to be near channels 250-300 indicated by “FZ” in the figure? — Okay, it seems you did. (5) I don’t expect a perfect match between Supp Fig. 7 and Figure 2g, but why is the signal stronger along B2 than B1 for the simulation while it is the opposite for the actual data (or are channel numbers completely different for Supp Fig. 7 than for Figure 2?).

Overall: I appreciate the observations (Figure 2) and the modeling attempts (Supp Figs 5-7). What I have issues with is that the two parts are not well connected and that a more complex model is strongly favored without convincing supporting argument. In terms of the timing and relative amplitude behavior, introducing the shallow low velocity layer does that. The topography and fault zone seem less important from Supp Fig. 5. The fault zone effect appears to be strong in Supp Fig.7 (and should be based on the strong contrast in the model L568-574). What part of the observations requires the topography (pointing to a feature in Fig. 2 and how topography in Supp Fig. 7 mimics that would help)? I don’t see (from Supp Fig. 7) why a moment tensor source would not work. The simulations do a good job showing that some complexity is required to match the observations qualitatively without the need to say which ones (that could be addressed in a more detailed modeling study), obviously a shallow velocity contrast plays a key role.

Below follow additional small suggestions for edits:

MAIN TEXT

L139. remove ‘fine’

L151-152. “B1 and B2 ...” — the next sentence explains that too, this sentence could be cut?

Figure 1. Yellow labels are difficult to read (particularly BN and VOR because of lighting/shading).

L161. EMOT or EMOV (as in figure)?

L174 “in reference¹⁰”?

L175 “until reaching the instruments”? or “wave to reach the instruments”

L183 “... (channels 315-340) in the proximity of a fault zone (FZ, Fig. 1).” — It is still not 100% clear which fault zone you are referring to since the fibre line in Figure 1 does not have labels. I assume it's the yellow dashed fault near/beneath geophone C671 assuming channel 1 is closest to the Pizzi Deneri Observatory and channels are about equally spaced. ... Okay, becomes clear from Figure 2g-h (but that's a bit convoluted)

L239-252 The seismic data (Fig. 3 b and e) also do not show the high-frequency peak; worthwhile mentioning?

L294-297 You could give some numbers considering the large potential range of scoriae thickness from the resonance frequency analysis (2.5 to 17 m, which is mainly due to the wide range of Vs given). Why not mention the MASW results in relation to the resonance frequency analysis.

Figure 4. The time axis in parts b and c has multiple “10:54:18”. Could you fix that (add decimal)?

L358 ‘being associated with tremor ...’?

L369 The data seem to end at 17:45 (and not 17:46 as the caption says)?

L376 “The according gamma ...” should this be “The corresponding gamma ...”?

L377 probably should explain “R” at first occurrence ... though R is an ‘event-per-time-rate’ rather than it's inverse? (see L703)

L380 What is “R . delta-t”, should be a multiplication sign?

Figure 4, what if you compared the LP events with swarm events rather than Southern California tectonic earthquakes?

L400 “DeGassing ...”

L407 “to sense” do you mean “to measure”?

L408 “understanding” does this refer to structural features and processes or to signals measured/sensed? Not clear.

L409 Do you mean “sensible” or “sensable”? In any case, not sure I'd use either word here.

L480 remove “and”?

L489-490. What do you mean? Not clear.

L490-491. Figure 7 though shows limits of the coupling?

L498 “seismometer”

L500 for the CMG3-ESPC you give the manufacturer, for the Trillium you do not. Why? Similarly for the geophones. Were the Guralps also buried only 25 cm?

L504 “Boise State University”

L530 “dx . du”? du/dx?

L570 “attenuation” or (L569-570) “quality factors $Q_p=100$ and $Q_s=80$ to account for attenuation, and ...”

L619 While ‘spectral’ the acronym MASW stands only for “Multichannel Analysis of Surface Waves”?

L718 “Codes to plot the figures” is relatively weak. That’s probably the least problem to re-do your analysis if anybody would chose to do so.

References. The styles used are slightly variable but I leave that up to the journal.

SUPPLEMENT

SL67 ‘and 1.5 s long’

SL70 ‘but for an isotropic ...’

Supp. Fig 8. Question: is the supplementary movie 3 still included (as the caption implies)? Based on the rebuttal letter it might not be part of the supplements any more - I’m a bit confused now (per section “Supplementary files” the video is still a part of the manuscript).

Supp. Fig 11. The thickness estimates for the scoriae (assuming the lowest interface represents the bottom of the layer) are about 20 m. How does this compare to the resonance frequency analysis (~3-17 m? L294-297 of main text)? It seems the text emphasizes similarity while not giving numbers (for the MASW results) that would indicate relatively high Vs (text part SL232-237) which constrain thicknesses better (or at least suggest the higher Vs and thus larger thicknesses)?

SL151 “In addition, lightning ...”

SL171 Please remove “3.8” in front of “0.4 km SE from Mount ...”, it’s a repeat of depth.
Supp Table 3. “Trillium”

SL224 “is the subject of many”

SL226 remove “all”

Point to point reply to the 2nd review

REVIEWER COMMENTS

Reviewer #2 (Remarks to the Author):

Second review of the manuscript “Fibre optic distributed acoustic sensing of volcanic events” by Jousset et al.

The revised version of the manuscript and supplementary material have improved with its new structure, so facilitating its reading. The authors answered the comments provided after the first review.

I however recommend to take into considerations the following comments for a further improvement. Page and line numbering refers to the version of the manuscript without correction tracking and comments are listed in order of appearance.

Page 2 Line 60: remove the term ‘injection’ and only use ‘transport’ of magma, as “injection” is frequently used in anthropic induced seismicity (e.g., exploitation of geothermal reservoirs).

We agree, done.

P3 L63: State of the volcano unrest -> state of the volcano or volcano unrest

Thanks. We have pushed this sentence and most of the paragraph into the supplementary note 1.

P3 L73: LP and VLP are not always observed on volcanoes, understanding their mechanism is important to understand the processes occurring within the volcanoes and these signals might be considered as further elements in assessing the state of the volcano.

Please rephrase, as VT and tremor are instead the signals commonly used for early warning. Or specify the particular case of Mt Etna.

Thanks for this important reminder. We have modified, although following remarks from reviewer 3 to make the text smoother we have moved most of the volcano seismology description in the supplementary information.

P3 L88: Rephrase the sentence in lines L88-L89, not clear in the context.

We have rephrased and moved into the supplementary note 1.

P4 L95: could you list the “fundamental eruption source parameters”?

We have added some of the important parameters in the sentence, focussing on the ones we address in the manuscript (new line 72).

P6 Fig1: please add B1 and B2 in legend. Another sketch in supplementary material could help the reader to better image the fiber configuration, as channel numbers are often used in the manuscript.

We agree, done. We have added a sketch as figure 1b.

P7: The videos were not available for this review. Be careful to insert them again.

Indeed, sorry for the confusion. The videos are the same as for the first version. We inserted them in this resubmission.

References are not inserted appropriately in this page, could you for instance rephrase in order to better understand the equations 5,6 and 7 of ref 10.

Rephrased. We have also found an error at (old) line 172: the call to reference 34 should be a call to 10 (new: ref 13).

P7 L175-176: sentence not clear: rephrase with something like to ‘In the DAS records, we analyse 2 sequences allowing us to determine the ground response and subsurface structure’.

Rephrased.

P7, L183: FZ is mentioned several times in manuscript, but is not on the map.

Done. We have also added fault 1 and fault 2 to identify them better in the text.

P10, L236: Fig 10 and 11 show velocities ranging from 200 to 600 m/s. Where did you detect V_s up to 1100m/s? can it be illustrated in supp material too? Please then ensure the deposit depth is then correctly estimated.

Supplementary Figure 10 (apparent velocities) shows actually apparent velocities ranging from 400 ms^{-1} to 1100 ms^{-1} and Supplementary Figure 11 (MASW) indeed between less than 200 ms^{-1} and 600 ms^{-1} . From the explosion, we record resonance frequencies ranging 16 Hz to 21 Hz. Considering the largest range of our estimated velocities (200 to 1100 ms^{-1}), the minimum and maximum thicknesses from surface (depth) are 2.5 m to 17 m. The key message is that all values are consistent and indicate that the infrasound waves can be used to probe subsurface of the ground. We limit ourselves to the “true” velocity values from MASW later in the paper, as we first resolve the wave-field and then could perform MASW analysis. Thicknesses which range 2.5 m (21 Hz, $V_s=200 \text{ ms}^{-1}$) until 4-5 m (16 Hz, 600 ms^{-1}). This would indicate that only the shallow portion of the ground/scoria layer would resonate. This discussion has been introduced after we have introduced the wavefield separation (new lines 280-300).

P10 L239: in a few lines you have the two different concepts resonance frequency and frequency resonance. Please rename one of them for better understanding.

Thanks to spot this. We changed the first one, as there is only one concept: the infrasound triggers the high frequencies, that we interpret as the resonance of the superficial layers of scoriae.

P16 from L346: part of references were shifted and have to be corrected as #32-> #33 and #33-> #34.

Indeed! Corrected.

P29 L613-615: This comment is more general on the estimations of velocities of propagation throughout the manuscript. With the different methods, you mention that lower velocities are observed in fault zones: could you please quantify?

It is quite challenging to quantify accurately velocities values in the faults, as faults zones are rather small in this context, and our methods use correlations of distant data points or dispersion curves which are disturbed by the faults. Possibly modelling would help understand how much velocity decrease would correspond to the modelled disturbance. However, as an illustration, we can look at Supplementary Figure 10 results. It appears that the lowest estimations are geographically corresponding to faults zone, whereas the highest estimations are for area far from fault zones. Therefore, we conclude that faults zones reveal lower relative velocities. A minimum value could be 200 ms^{-1} .

In DAS strain rate and strain validation, you obtain c_x varying between 400 and 1110m/s mentioning that are 'upper limit values of the true propagation velocity'. Could you explain?

The explanation is given at the lines 615-616 (old manuscript), 630-633 (new manuscript), using the equation $\tau_s = \frac{d_s}{V_h} \cos \gamma$. γ is the angle between the incident wave and the cable. If γ is zero, then the waves are propagating along the cable and V_h will be minimal. If γ increases, then the apparent velocity increases. Therefore, the estimated values are upper limits of the true velocity, which can be obtained only for waves propagating along the cable, which is only partially the case. This is also verified in the MASW results, where we find max $V_s \sim 600 \text{ m/s}$.

P30 L630 This sentence is not very clear. For the inversion, do you parametrize the model space with layer thicknesses and shear wave velocities? The a posteriori probability function is a function of the model parameters, so its sampling should refer to thickness and shear wave velocities (and not phase velocities that are your observations).

This is exact. We have modified the text accordingly, replacing phase velocity by shear-wave velocity (line 630).

Reviewer #3 (Remarks to the Author):

Review of revised manuscript "Fibre optic distributed acoustic sensing of volcanic events" by Jousset et al. for publication in Nature Communications

First, I appreciate the detailed replies to my initial comments and the attempts to incorporate some of the suggestions. Thanks for highlighting some of the key features in the discussion/conclusions; providing such add'l information/explanation of the advantages of DAS is what I was hoping to see. I think this strengthens the arguments 'pro DAS'.

I have three remaining/new issues:

(1) I really do not see the point of adding a long intro about 'volcano seismology' (L56-L107). The writing is a bit repetitive (L61-63 and L72-74) and the introduction is now more difficult/convoluted to read (for me). The entire purpose of this summary — it reads like a review of phenomena — is not clear. Please edit and relate these features more directly to contributions from DAS (focus). I understand the add-ons are in response to reviewer 1 but they are so general that it's difficult to see the relevance for the manuscript. The introduction now seems weaker and less to the point. The point is you are introducing DAS to volcano monitoring and here give some examples to show its potential and usefulness.

We have reduced the length of volcano seismology and pushed the lengthy description into the additional supplementary note 1. It suits certainly better there, to allow non-volcano-seismologists to acquire background information, without diverting the main message in the introduction. In the introduction, we have focussed on the essential elements that we address in the manuscript (explosion, LP and tremor) and to which DAS brings elements to better understand them.

(2) The text has been changed significantly in several places (particularly in the introduction and discussion parts). The style is now less smooth than the initial version and I would encourage the authors to go through the text with that (smoothness of reading) in mind. I do not have a specific example, sorry, but the parts that were changed (and/or are new) are usually more difficult to follow than before.

We have read again and smoothed, by comparing with the initial version 1.

(3) I appreciate the reply about the wave field simulation. However,
(3.a) I do not see a point in the particle motion plots. They are shown for simulations but not for actual data. How are readers supposed to know/appreciate that more complicated models produce particle motions that 'better' follow the actual data when there's no example from the actual data? I do not see a reason to have Supp. Fig 6. considering current text and other visuals.

Agreed. We have refurbished complete the waveform modelling. We introduced data for waveform, particle motions and strain rate. We added systematic computations for all cases combination out of the 3 parameters introduced (topography, fault zone, velocity models), making now 8 possible combinations. Then comparison between cases and data can be assessed better. In addition, instead of only a shallow velocity model, we have introduced the velocity models coming from travel time tomography, which makes the velocity model closer to the true one. We observe in that case that indeed the amplitudes on the two branches are similar, suggesting that the velocity model is responsible of the relative change of amplitudes due to a

wavefield rotation. Further analysis on those aspects is out the scope of this study, as the objective here it to illustrate the effect of those elements on the strain rate amplitudes.

(3.b) Need for topography, fault zone, shallow low velocities as postulated. Supp. Fig. 5 b and c are basically identical (except for amplitude). That means the effect of the fault zone is negligible? Why not? Supp. Fig. 7 implies that the fault zone has a large effect on the strain rate as the results for cases b (topography) and c (topo + fault zone) look vastly different in the fault zone (FZ) region. How is that possible if the traces (Supp. Fig 5 b and c) in the FZ region are very similar? This is not at all clear.

Strain rate is much more sensitive to small lateral changes in ground velocity than particle motion and waveforms themselves. Strain rate addresses changes of the velocity gradient with time. We have enhanced this observation plotting data for several geophones: in the supplementary figure 5 we have plotted waveforms for C670, C671 and C672. Waveforms are rather similar, although the strain rate (Fig. 2, and also now supplementary figure 7) clearly shows a strong change in amplitudes in the observed strain rate. Our modelling attempts reflect clearly the FZ features.

(3.c) L183-187 of the main text, “Using 3D wave propagation we demonstrate that ... topography and local volcanic structures explain the wave-field variability of travel times, ... particle motions and strain rate amplitudes”. I don't see how the modeling demonstrates this.

Agreed. We have replaced demonstrates by illustrates.

Are you expecting the reader to translate Figure 2g into time series or simply compare Figure 2b with the Supp Fig. 5 traces? Or are you suggesting that Supp. Fig 7e looks the most similar to the actual data (Fig. 2) from all the model runs? This is not clear. And which parts of Fig. 2 are close to Supp, Fig. 7?

We have introduced data in all supplementary figure 5, 6 and 7, so that models shown can be better compared with observations.

Would the traces of geophone C671 located in the fault zone (instead of CAZG as shown in Fig. 2 b) resemble the simulation results (it should somewhat, based on the text)? Why not show C671?

Agreed. We have enhanced this observation using geophone data. In the supplementary figure 5 we have plotted waveforms for C670, C671 and C672.

(3.d) Supp Fig 7. (SF7)

(i) I don't understand the caption. Cases e and f produces similarly 'complex' waveforms (Supp Fig. 5) and have similar overall timing. For a given source strength, amplitudes for cases e and f differ, but why not simply make the source stronger for f to make the signal larger (what is the argument against that?).

From the infrasound signal, we have estimated the energy of the explosion at the source to be $2.5 \cdot 10^{11}$ J. This corresponds to a pressure of about 0.4-0.8 MPa at the frequencies of interest (0.1-5 Hz). In the case of a single force in our modelling, this pressure is applied to a horizontal surface of 50x50 m (grid spatial sampling of 50m) with corresponds to a force with magnitude $1 \sim 2 \cdot 10^9$ N. The synthetic amplitudes at the virtual seismic stations (velocities) and at the virtual fibre (starin rate) match amplitudes of observations. When the source comprises 3 perpendicular dipole (modelling an explosion), synthetic amplitudes require a pressure amplitude much larger than the one derived from the infrasound signals to match observed amplitudes (strain rate and velocities). In the literature, there are many cases where vertical single force has been shown to be more adapted that explosive sources. A downward force is required to compensate for the upward momentum of the volcanic ejecta (e.g. Ohminato et al., 2004). Therefore, we have removed the explosion as a potential source in the shown models, rather we provide a short explanation.

(ii) What do you mean by “more than 2 seconds”? Supp Fig. 5 clearly shows arrivals < 2 s for all examples (including preferred case f) and more specifically there’s no difference in timing between cases d-f?

We have improved the explanation. It was indeed not as clear as we wanted! ~ 2 s is the time difference between the travel time of wave arrival in observed and synthetic cases. Our modelling indicates that used velocities may be too fast, as the main waves arrives much faster than their counterparts in the data (maximum amplitudes for example).

(iii) Looking at Supp Fig. 7 (and ignoring the absolute amplitudes), cases d, e, and f look very much alike. That means that the most important feature for the modeling exercise is the introduction of the shallow low velocity structure. Why not acknowledge that?

We agree. We have made clearer what each of the three elements chosen (topography, fault zone and tomographic models) bring. 1. The fault zone has its importance to explain larger amplitudes at the location of the fault, where energy is trapped: in correspondence to the fault zone, DAS observations show a slower velocity (Supplementary Fig. 10) and amplification of the signals (Fig. 2). The tomography contribute largely to make the wavefield rotate and the topography adjust the amplitudes better, although indeed not perfectly.

(iv) I am also surprised about the strong signal in the ~ 250 -300 channel range considering that the fault zone is near C671 that is near channel 340 or so; or did you move the fault zone to be near channels 250-300 indicated by “FZ” in the figure? — Okay, it seems you did.

In the previous model used, we have indeed introduced the fault zone at a reasonable distance from the branch B2, as the resolution of the model is only 50 m. We have re computed all synthetic by decreasing the grid nodes of the modelling to 30 m. This allows to locate the fault zone better, and place it a reasonably close to the reality.

(5) I don’t expect a perfect match between Supp Fig. 7 and Figure 2g, but why is the signal stronger along B2 than B1 for the simulation while it is the opposite for the actual data (or are channel numbers completely different for Supp Fig. 7 than for Figure 2?).

The amplitudes in branch B1 are indeed larger than in branch B2: this is a feature of the field at that location, as already observed previously for seismometers. We performed another field experiment in 2019 with a different cable layout and characteristics (Currenti et al, 2019, reference 44), and the same features occur: amplitudes in the east/west direction are larger than in the north/south. This is not explained so far.

Our modelling attempts illustrate indeed that combining topography and tomographic models on which a shallow velocity layer (in our model: 100 m/1700 ms⁻¹) is added allows to match observations better and understand those features. Whereas the models without topography and fault zones do not explain this feature at all, the only model where we have this is indeed the one where we have all complexities involved, demonstrating (illustrating) that all elements are indeed relevant: 1. Tomographic models with shallow low velocity explain travel times and amplitudes 2. Fault zone explains larger amplitudes locally and topography allows a better relative amplitude between B1 and B2, which suggests that the velocity distribution in the plumbing system makes the wavefield rotate at Pino delle concazze.

Overall: I appreciate the observations (Figure 2) and the modeling attempts (Supp Figs 5-7). What I have issues with is that the two parts are not well connected and that a more complex model is strongly favored without convincing supporting argument. In terms of the timing and relative amplitude behavior, introducing the shallow low velocity layer does that. The topography and fault zone seem less important from Supp Fig. 5. The fault zone effect appears to be strong in Supp Fig.7 (and should be based on the strong contrast in the model L568-574).

We have better connected to two, by introducing data in the final figure, and we have introduced a larger number of cases to focus on illustrating how elements contribute to trying to explain observed features or not.

What part of the observations requires the topography (pointing to a feature in Fig. 2 and how topography in Supp Fig. 7 mimics that would help)?

There is certainly no single feature in the data that can be identified and isolated to demonstrate in this particular case that topography has in our example a visible effect. However, when comparing model g and h, particle motion appear slightly flatter with topography and relative strain rate amplitudes between B1 and B2 are closer to data than with other models.

I don't see (from Supp Fig. 7) why a moment tensor source would not work.

Due to its very close proximity to the surface, the single vertical force is the most adapted source model. The ejection of material in the atmosphere, implies that a counter force downward keeps equilibrium in the system. In addition, the source amplitude is derived using the infrasound measurement. Using the magnitude obtained from the infrasound data, on modelling reveal that an explosion source does not explain the amplitudes. If we require an explosion as source model to match the amplitudes of the seismic observations, then the pressure required is much larger and does not match the amplitude in the infrasound record.

The simulations do a good job showing that some complexity is required to match the observations qualitatively without the need to say which ones (that could be addressed in a more detailed modeling study), obviously a shallow velocity contrast plays a key role.

We have modified the text to match those requirements.

Below follow additional small suggestions for edits:

MAIN TEXT

L139. remove 'fine'

Done

L151-152. "B1 and B2 ..." — the next sentence explains that too, this sentence could be cut?

We have changed the presentation, and introduced a new sketch.

Figure 1. Yellow labels are difficult to read (particularly BN and VOR because of lighting/shading).

Done.

L161. EMOT or EMOV (as in figure)?

Changed in figure. (typo).

L174 "in reference¹⁰"?

Modified.

L175 "until reaching the instruments"? or "wave to reach the instruments"

Changed.

L183 "... (channels 315-340) in the proximity of a fault zone (FZ, Fig. 1)." — It is still not 100% clear which fault zone you are referring to since the fibre line in Figure 1 does not have labels. I assume it's the yellow dashed fault near/beneath geophone C671 assuming channel 1 is closest to the Pizzi Deneri Observatory and channels are about equally spaced. ... Okay, becomes clear from Figure 2g-h (but that's a bit convoluted)

Following suggestion of reviewer 2, we have added a sketch of the deployment, in association to the lines 152-158, which are now moved to the caption Fig. 1b.

L239-252 The seismic data (Fig. 3 b and e) also do not show the high-frequency peak; worthwhile mentioning?

Thanks for valuable suggestion. Added.

L294-297 You could give some numbers considering the large potential range of scoriae thickness from the resonance frequency analysis (2.5 to 17 m, which is mainly due to the wide range of Vs given). Why not mention the MASW results in relation to the resonance frequency analysis.

We have rephrased a bit to mention clearer the relation between the explosion result and the MASW results. Both are mentioned, but their relationship was not clear enough indeed.

Figure 4. The time axis in parts b and c has multiple “10:54:18”. Could you fix that (add decimal)?

Fixed.

L358 ‘being associated with tremor ...’?

We have rephrased this sentence.

L369 The data seem to end at 17:45 (and not 17:46 as the caption says)?

Corrected.

L376 “The according gamma ...” should this be “The corresponding gamma ...”?

Corrected.

L377 probably should explain “R” at first occurrence ... though R is an ‘event-per-time-rate’ rather than it’s inverse? (see L703)

Corrected.

L380 What is “R . delta-t”, should be a multiplication sign?

Corrected.

Figure 4, what if you compared the LP events with swarm events rather than Southern California tectonic earthquakes?

The reviewer means certainly Figure 5. This is a very relevant suggestion. However, during our measurement, no volcano tectonic VT swarm occurred, making the comparison impossible for our events. Nevertheless, in reference 32 data during eruption and quiet period has been compared at Etna (see reference 32). While during quiet period Etna seismicity follow a tectonic model (like Vesuvius quite volcano actually), the inter event time distribution follow a different law. This is already indicated in our manuscript at lines 703-705. In order to make it clearer, we

have introduced the description of the method in the text, allowing this point to be clarified, and also mentioning that although Etna was quiet, our events follow distribution similar to LP ones.

L400 “DeGassing ...”

Corrected.

L407 “to sense” do you mean “to measure”?

Corrected.

L408 “understanding” does this refer to structural features and processes or to signals measured/sensed? Not clear.

Rephrased.

L409 Do you mean “sensible” or “sensable”? In any case, not sure I’d use either word here.

Corrected, removed.

L480 remove “and”?

Corrected.

L489-490. What do you mean? Not clear.

Corrected, sentence removed, as it does not bring anything.

L490-491. Figure 7 though shows limits of the coupling?

The limitation has been introduced (wind).

L498 “seismometer”

Corrected.

L500 for the CMG3-ESPC you give the manufacturer, for the Trillium you do not. Why? Similarly for the geophones. Were the Guralps also buried only 25 cm?

Corrected: we added the company name for all sensors. The depth of the sensors is given at line 501. The guralps were located slightly deeper.

L504 “Boise State University”

Corrected.

L530 “dx . du”? du/dx?

The is the scalar product, explanation introduced now.

L570 “attenuation” or (L569-570) “quality factors $Q_p=100$ and $Q_s=80$ to account for attenuation, and ...”

Corrected.

L619 While ‘spectral’ the acronym MASW stands only for “Multichannel Analysis of Surface Waves”?

Corrected.

L718 “Codes to plot the figures” is relatively weak. That’s probably the least problem to re-do your analysis if anybody would chose to do so.

Corrected.

References. The styles used are slightly variable but I leave that up to the journal.

We checked found indeed issues and corrected, however there may be some typos left.

SUPPLEMENT

SL67 ‘and 1.5 s long’

Corrected

SL70 ‘but for an isotropic ...’

Corrected.

Supp. Fig 8. Question: is the supplementary movie 3 still included (as the caption implies)? Based on the rebuttal letter it might not be part of the supplements any more - I’m a bit confused now (per section “Supplementary files” the video is still a part of the manuscript).

There are different opinions on whether to leave this movie or not. We decided finally to leave it and corrected the text accordingly.

Supp. Fig 11. The thickness estimates for the scoriae (assuming the lowest interface represents the bottom of the layer) are about 20 m. How does this compare to the resonance frequency analysis (~3-17 m? L294-297 of main text)? It seems the text emphasizes similarity while not giving numbers (for the MASW results) that would indicate relatively high Vs (text part SL232-237) which constrain thicknesses better (or at least suggest the higher Vs and thus larger thicknesses)?

We have included more discussion in the text concerning those results and the explosion triggered resonance. Indeed, using the result of the MASW, we can infer which part of the subsurface resonated. The whole layers imaged here may not have all resonate.

SL151 “In addition, lightning ...”

Corrected.

SL171 Please remove “3.8” in front of “0.4 km SE from Mount ...”, it’s a repeat of depth. Supp Table 3. “Trillium”

Both corrected.

SL224 “is the subject of many”

Corrected.

SL226 remove “all”

Corrected.

REVIEWER COMMENTS

Reviewer #3 (Remarks to the Author):

Review of revised manuscript “Fibre optic distributed acoustic sensing of volcanic events” by Jousset et al. for publication in Nature Communications

I have no further issues with the manuscript. The authors addressed all of my concerns. The text reads very well now. Moving the section about volcano seismology to the supplement is perfect because it is useful, compact information for non-experts but does not take up time for experts who can skip it. I very much appreciate the effort to change the wave field simulation part. The presentation is now much clearer and the words directly address how each complexity affects the wave field and why it is required by the data.

Two tiny things:

L566. Please check right side of equation (1). Numerator is 0 because of a missing “-“ sign?

SuppL88. ‘sensitivity is higher ...’?

Point to point response to the reviewers' comments.

REVIEWER COMMENTS

Reviewer #3 (Remarks to the Author):

Review of revised manuscript "Fibre optic distributed acoustic sensing of volcanic events" by Jousset et al. for publication in Nature Communications

I have no further issues with the manuscript. The authors addressed all of my concerns. The text reads very well now. Moving the section about volcano seismology to the supplement is perfect because it is useful, compact information for non-experts but does not take up time for experts who can skip it. I very much appreciate the effort to change the wave field simulation part. The presentation is now much clearer and the words directly address how each complexity affects the wave field and why it is required by the data.

Thank you very much for those positive comments.

Two tiny things:

L566. Please check right side of equation (1). Numerator is 0 because of a missing "-" sign?

The "-" sign has been checked. It was already present.

SuppL88. 'sensitivity is higher ...'?

Corrected, thanks!